# Towards Consistency in Adversarial Classification

**Laurent Meunier**[1,2]   **Raphaël Ettedgui**[1]   **Rafael Pinot**[3]
**Yann Chevaleyre**[1]   **Jamal Atif**[1]

[1]Université Paris-Dauphine, PSL Research University, CNRS, LAMSADE, Paris, France
[2]Meta AI Research, Paris, France
[3] École Polytechnique Fédérale de Lausanne (EPFL), Switzerland

## Abstract

In this paper, we study the problem of consistency in the context of adversarial examples. Specifically, we tackle the following question:

*Can surrogate losses still be used as a proxy for minimizing the $0/1$ loss in the presence of an adversary that alters the inputs at test-time?*

Different from the standard classification task, this question cannot be reduced to a point-wise minimization problem, and calibration needs not to be sufficient to ensure consistency. In this paper, we expose some pathological behaviors specific to the adversarial problem, and show that no convex surrogate loss can be consistent or calibrated in this context. It is therefore necessary to design another class of surrogate functions that can be used to solve the adversarial consistency issue. As a first step towards designing such a class, we identify sufficient and necessary conditions for a surrogate loss to be calibrated in both the adversarial and standard settings. Finally, we give some directions for building a class of losses that could be consistent in the adversarial framework.

## 1 Introduction

State-of-the-art machine learning classifiers are known to be vulnerable to adversarial example attacks [9, 18], i.e., perturbation of the input data at test time that, while imperceptible, can significantly influence the classifier's output. This can have extreme consequences in real-life scenarios such as autonomous cars [19]. Therefore, it is necessary to design *robust classifiers* that present worst-case guarantees against a range of possible perturbations. To account for the possibility of an adversary manipulating the inputs at test time, we need to revisit the standard risk minimization problem by penalizing any classification model that might change its decision when the point of interest is slightly changed. Essentially, this is done by replacing the standard (pointwise) $0/1$ loss with an adversarial version that mimics its behavior locally but also penalizes any error in a given region around the point on which it is evaluated.

Yet, just like the $0/1$ loss, its adversarial counterpart is not convex, which renders the risk minimization difficult. To circumvent this limitation, we take inspiration from the standard learning theory approach which consists in solving a simpler optimization problem where the non-convex loss function is replaced by a convex surrogate. In general, the surrogate loss is chosen to have a property called *consistency* [20, 6, 17], which essentially guarantees that any sequence of classifiers that minimizes the surrogate objective must also be a sequence that minimizes the Bayes risk. In the context of standard classification, a large family of convex losses, called *classifier-consistent*, exhibits this property. This class notoriously includes the hinge loss, the logistic loss and the square loss.

However, the adversarial version of these surrogate losses need not to exhibit the same consistency properties with respect to the adversarial $0/1$ loss. In fact, most existing results in the standard

36th Conference on Neural Information Processing Systems (NeurIPS 2022).

framework rely on a reduction of the global consistency problem to a point-wise problem, called *calibration*. However, this approach is not feasible in the adversarial setting, because the new losses are by nature non-point-wise: the optimum for a given input may depend on yet a whole other set of inputs [1, 3]. Studying the concepts of calibration and consistency in an adversarial context remains an open and understudied issue. Furthermore, this is a complex and technical area of research, that requires a rigorous analysis, since small tweaks in definitions can quickly make results meaningless or inaccurate. This difficulty is illustrated in the literature, where articles published in high profile conferences tend to contradict or refute each other [4, 1, 3].

**Objective & Contributions.** The objective of our work is to try and identify possible sources of confusion that may hinder the understanding of the concepts of calibration and consistency in the adversarial setting. In particular, we first come back in Section 2 on the problem of consistency and calibration in the standard setting and carefully define their adversarial counterpart. In doing so, we note that previous papers studying adversarial surrogate losses [4, 1, 3] did not use the same $0/1$-loss as the one used in seminal papers on consistency [20, 6, 17]. This difference might be crucial since it may lead to inaccurate results (see Section 5). We then study in Section 3, the problem of calibration in the adversarial setting and provide both necessary and sufficient conditions for a loss to be calibrated in this setting. It also worth noting that our results are easily extendable to $\mathcal{H}$-calibration (see Appendix L). One on the main takeaway of our analysis is that no convex surrogate loss can be calibrated in the adversarial setting. We however characterize a set of non-convex loss functions, namely *shifted odd functions* that solve the calibration problem in the adversarial setting. Finally, we focus on the problem of consistency in the adversarial setting in Section 4. Based on min-max arguments, we provide insights that might help paving a way to prove consistency of shifted odd functions in the adversarial setting. Specifically, we prove strong duality results for these losses and show tight links with the $0/1$-loss. From these insights, we are able to provide a close but weaker property to consistency.

## 2 Notions of Calibration and Consistency

Let us consider a classification task with input space $\mathcal{X}$ and output space $\mathcal{Y} = \{-1, +1\}$. Let $(\mathcal{X}, d)$ be a proper Polish (i.e. completely separable) metric space representing the inputs space. For all $x \in \mathcal{X}$ and $\delta > 0$, we denote $B_\delta(x)$ the closed ball of radius $\delta$ and center $x$. We also assume that for all $x \in \mathcal{X}$ and $\delta > 0$, $B_\delta(x)$ contains at least two points[1]. Let us also endow $\mathcal{Y}$ with the trivial metric $d'(y, y') = \mathbf{1}_{y \neq y'}$. Then the space $(\mathcal{X} \times \mathcal{Y}, d \oplus d')$ is a proper Polish space. For any Polish space $\mathcal{Z}$, we denote $\mathcal{M}_+^1(\mathcal{Z})$ the Polish space of Borel probability measures on $\mathcal{Z}$. We will denote $\mathcal{F}(\mathcal{Z})$ the space of real valued Borel measurable functions on $\mathcal{Z}$. Finally, we denote $\bar{\mathbb{R}} := \mathbb{R} \cup \{\infty, +\infty\}$.

### 2.1 Notations and Preliminaries

The $0/1$-loss is both non-continuous and non-convex, and its direct minimization is a difficult problem. The concepts of calibration and consistency aim at identifying the properties that a loss must satisfy in order to be a good surrogate for the minimization of the $0/1$-loss. In this section, we define these two concepts and explain the difference between them. First of all, we need to give a general definition of a loss function.

**Definition 2.1** (Loss function). *A loss function is a function $L : \mathcal{X} \times \mathcal{Y} \times \mathcal{F}(\mathcal{X}) \to \mathbb{R}$ such that $L(\cdot, \cdot, f)$ is measurable for all $f \in \mathcal{F}(\mathcal{X})$.*

Note that this definition is not specific to the standard or adversarial case. In general, the loss at point $(x, y)$ can either depend only on $f(x)$, or on other points related to $x$ (e.g. the set of points within a distance $\varepsilon$ of $x$). We now recall the definition of the risk associated with a loss $L$ and a distribution $\mathbb{P}$.

**Definition 2.2** (*L*-risk of a classifier). *For a given loss function L, and a Borel probability distribution $\mathbb{P}$ over $\mathcal{X} \times \mathcal{Y}$ we define the risk of a classifier $f$ associated with the loss L and a distribution $\mathbb{P}$ as*

$$\mathcal{R}_{L,\mathbb{P}}(f) := \mathbb{E}_{(x,y)\sim\mathbb{P}}\left[L(x, y, f)\right].$$

*We also define the optimal risk associated with the loss L as*

$$\mathcal{R}_{L,\mathbb{P}}^\star := \inf_{f\in\mathcal{F}(\mathcal{X})} \mathcal{R}_{L,\mathbb{P}}(f)$$

---

[1]For instance, for any norm $\|\cdot\|$, $(\mathbb{R}^d, \|\cdot\|)$ is a Polish metric space satisfying this property.

Essentially, the risk of a classifier is defined as the average loss over the distribution $\mathbb{P}$. When the loss $L$ is difficult to optimize in practice (e.g when it is non-convex or non-differentiable), it is often preferred to optimize a surrogate loss function instead. In the literature [20, 6, 17], the notion of surrogate losses has been studied as a consistency problem. In a nutshell, a surrogate loss is said to be consistent if any minimizing sequence of classifiers for the risk associated with the surrogate loss is also one for the risk associated with $L$. Formally, the notion of consistency is as follows.

**Definition 2.3** (Consistency). *Let $L_1$ and $L_2$ be two loss functions. For a given $\mathbb{P} \in \mathcal{M}_1^+(\mathcal{X} \times \mathcal{Y})$, $L_2$ is said to be consistent for $\mathbb{P}$ with respect to $L_1$ if for all sequences $(f_n)_n \in \mathcal{F}(\mathcal{X})^{\mathbb{N}}$ :*

$$\mathcal{R}_{L_2,\mathbb{P}}(f_n) \to \mathcal{R}_{L_2,\mathbb{P}}^{\star} \implies \mathcal{R}_{L_1,\mathbb{P}}(f_n) \to \mathcal{R}_{L_1,\mathbb{P}}^{\star} \tag{1}$$

*Furthermore, $L_2$ is said consistent with respect to a loss $L_1$ the above holds for any distribution $\mathbb{P}$.*

Consistency is in general a difficult problem to study because of its high dependency on the distribution $\mathbb{P}$ at hand. Accordingly, several previous works [20, 5, 17] introduced a weaker notion to study a pointwise version consistency. This simplified notion is called *calibration* and corresponds to consistency when $\mathbb{P}$ is a combination of Dirac distributions. The main building block in the analysis of the calibration problem is the calibration function, defined as follows.

**Definition 2.4** (Calibration function). *Let $L$ be a loss function. The calibration function $\mathcal{C}_L$ is*

$$\mathcal{C}_L(x, \eta, f) := \eta L(x, 1, f) + (1 - \eta) L(x, -1, f),$$

*for any $\eta \in [0, 1]$, $x \in \mathcal{X}$ and $f \in \mathcal{F}(\mathcal{X})$. We also define the optimal calibration function as*

$$\mathcal{C}_L^{\star}(x, \eta) := \inf_{f \in \mathcal{F}(\mathcal{X})} \mathcal{C}_L(x, \eta, f).$$

Note that for any $x \in \mathcal{X}$ and $\eta \in [0, 1]$, when $\mathbb{P} = \eta \delta_{(x,+1)} + (1 - \eta) \delta_{(x,-1)}$, we $\mathcal{C}_L(x, \eta, f) = \mathcal{R}_{L,\mathbb{P}}(f)$ with The calibration function thus corresponds to then a pointwise notion of the risk, evaluated at point $x$. $\eta$ corresponds in this case to the conditional probability of $y = 1$ given $x$. We now define the calibration property of a surrogate loss.

**Definition 2.5** (Calibration). *Let $L_1$ and $L_2$ be two loss functions. We say that $L_2$ is* calibrated *with regards to $L_1$ if for every $\xi > 0$, $\eta \in [0, 1]$ and $x \in \mathcal{X}$, there exists $\delta > 0$ such that for all $f \in \mathcal{F}(\mathcal{X})$,*

$$\mathcal{C}_{L_2}(x, \eta, f) - \mathcal{C}_{L_2}^{\star}(x, \eta) \leq \delta \implies \mathcal{C}_{L_1}(x, \eta, f) - \mathcal{C}_{L_1}^{\star}(x, \eta) \leq \xi.$$

*Furthermore, we say that $L_2$ is* uniformly calibrated *with regards to $L_1$ if for every $\xi > 0$, there exists $\delta > 0$ such that for all $\eta \in [0, 1]$, $x \in \mathcal{X}$ and $f \in \mathcal{F}(\mathcal{X})$ we have*

$$\mathcal{C}_{L_2}(x, \eta, f) - \mathcal{C}_{L_2}^{\star}(x, \eta) \leq \delta \implies \mathcal{C}_{L_1}(x, \eta, f) - \mathcal{C}_{L_1}^{\star}(x, \eta) \leq \xi.$$

**Connection between calibration and consistency.** It is always true that calibration is a necessary condition for consistency. Yet there is no reason, in general, for the converse to be true. However, in the specific context usually studied in the literature (i.e., the standard classification with a well-defined $0/1$-loss), the notions of consistency and calibration have been shown to be equivalent. [20, 6, 17]. In the next section, we come back on existing results regarding calibration and consistency in this specific (standard) classification setting.

## 2.2 Existing Results in the Standard Classification Setting

Classification is a standard task in machine learning that consists in finding a classification function $h : \mathcal{X} \to \mathcal{Y}$ that maps an input $x$ to a label $y$. In binary classification, $h$ is often defined as the sign of a real valued function $f \in \mathcal{F}(\mathcal{X})$. The loss usually used to characterize classification tasks corresponds to the accuracy of the classifier $h$. When $h$ is defined as above, this loss is defined as follows.

**Definition 2.6** ($0/1$ loss). *Let $f \in \mathcal{F}(\mathcal{X})$. We define the $0/1$ loss as follows*

$$l_{0/1}(x, y, f) = \mathbf{1}_{y \times sign(f(x)) \leq 0}$$

*with a convention for the sign, e.g. $sign(0) = 1$. We will denote $\mathcal{R}_{\mathbb{P}}(f) := \mathcal{R}_{l_{0/1},\mathbb{P}}(f)$, $\mathcal{R}_{\mathbb{P}}^{\star} := \mathcal{R}_{l_{0/1},\mathbb{P}}^{\star}$, $\mathcal{C}(x, \eta, f) := \mathcal{C}_{l_{0/1}}(x, \eta, f)$ and $\mathcal{C}^{\star}(x, \eta) := \mathcal{C}_{l_{0/1}}^{\star}(x, \eta)$.*

Note that this 0/1-loss is different from the one introduced by Bao et al. [4], Awasthi et al. [1, 3]: they used $\mathbf{1}_{y \times f(x) \leq 0}$ which is a usual 0/1 loss but unadapted to consistency and calibrated study (see Section 5 for details). Some of the most prominent works [20, 6, 17] among them focus on the concept of margin losses, as defined below.

**Definition 2.7** (Margin loss). *A loss $L_\phi$ is said to be a* margin loss *if there exists a measurable function $\phi : \mathbb{R} \to \mathbb{R}_+$ such that:*

$$L_\phi(x, y, f) = \phi(yf(x))$$

For simplicity, we will say that $\phi$ is a margin loss function and we will denote $\mathcal{R}_\phi$ and $\mathcal{C}_\phi$ the risk associated with the margin loss $\phi$. Notably, it has been demonstrated in several previous works [20, 6, 17] that, for a margin loss $\phi$, we have always have $\mathcal{C}_\phi^\star(x, \eta) = \inf_{\alpha \in \mathbb{R}} \eta \phi(\alpha) + (1 - \eta)\phi(-\alpha)$. This is in particular one of the main observation allowing to show the following strong result about the connection between consistency and calibration.

**Theorem 2.1** (Zhang [20], Bartlett et al. [6], Steinwart [17]). *Let $\phi : \mathbb{R} \to \mathbb{R}_+$ be a continuous margin loss. Then the three following assertions are equivalent: (i) $\phi$ is calibrated with regards to $l_{0/1}$, (ii) $\phi$ is uniformly calibrated $l_{0/1}$, (iii) $\phi$ is consistent with regards to $l_{0/1}$.*

*Moreover, if $\phi$ is convex and differentiable at $0$, then $\phi$ is calibrated if and only $\phi'(0) < 0$.*

The Hinge loss $\phi(t) = \max(1 - t, 0)$ and the logistic loss $\phi(t) = \log(1 + e^{-t})$ are classical examples of convex consistent losses. Convexity is a desirable property for faster optimization of the loss, but there exist other non-convex losses that are calibrated as the ramp loss ($\phi(t) = \frac{1}{2}(\max(1 - t, 0) + \max(-1 - t, 0))$) or the sigmoid loss ($\phi(t) = (1 + e^t)^{-1}$). These losses are plotted on the left in Figure 1. In the next section, we present the adversarial classification setting for which Theorem 2.1 may not hold anymore.

**Remark 1.** *The equivalence between calibration and consistency is a consequence from the fact that, over the large space of measurable functions, minimizing the loss pointwisely in the input by desintegrating with regards to $x$ is equivalent to minimize the whole risk over measurable functions. This result is very powerful and simplify the study of calibration in the standard setting.*

## 2.3 Calibration and Consistency in the Adversarial Setting.

We now consider the adversarial classification setting where an adversary tries to manipulate the inputs at test time. Given $\varepsilon > 0$, they can move each point $x \sim \mathbb{P}$ to another point $x'$ which is at distance at most $\varepsilon$ from $x$[2]. The goal of this adversary is to maximize the 0/1 risk the shifted points from $\mathbb{P}$. Formally, the loss associated to adversarial classification is defined as follows.

**Definition 2.8** (Adversarial 0/1 loss). *Let $\varepsilon \geq 0$. We define the adversarial $0/1$ loss of level $\varepsilon$ as:*

$$l_{0/1,\varepsilon}(x, y, f) = \sup_{x' \in B_\varepsilon(x)} \mathbf{1}_{y \, \text{sign}(f(x)) \leq 0}$$

*We will denote $\mathcal{R}_{\varepsilon, \mathbb{P}}(f) := \mathcal{R}_{l_{0/1,\varepsilon}, \mathbb{P}}(f)$, $\mathcal{R}_{\varepsilon, \mathbb{P}}^\star := \mathcal{R}_{l_{0/1,\varepsilon}, \mathbb{P}}^\star$, $\mathcal{C}_\varepsilon(x, \eta, f) := \mathcal{C}_{l_{0/1,\varepsilon}}(x, \eta, f)$ and $\mathcal{C}_\varepsilon^\star(x, \eta) := \mathcal{C}_{l_{0/1,\varepsilon}}^\star(x, \eta)$ for every $\mathbb{P}$, $x$, $f$ and $\eta$.*

**Specificity of the adversarial case** The adversarial risk minimization problem is much more challenging than its standard counterpart because an inner supremum is added to the optimization objective. With this inner supremum, it is no longer possible to reduce the distributional problem to a pointwise minimization as it is usually done in the standard classification framework. In fact, the notions of consistency and calibration are significantly different in the adversarial setting. This means that the results obtained in the standard classification may no longer be valid in the adversarial setting (e.g., the calibration need not be sufficient for consistency), which makes the study of consistency much more complicated. As a first step towards analyzing the adversarial classification problem, we now adapt the notion of margin loss to the adversarial setting.

**Definition 2.9** (Adversarial margin loss). *Let $\phi : \mathbb{R} \to \mathbb{R}_+$ be a margin loss and $\varepsilon \geq 0$. We define the adversarial loss of level $\varepsilon$ associated with $\phi$ as:*

$$\phi_\varepsilon(x, y, f) = \sup_{x' \in B_\varepsilon(x)} \phi(yf(x'))$$

---

[2]Note that after shifting $x$ to $x'$, the point need not be in the support of $\mathbb{P}$ anymore.

*We say that $\phi$ is adversarially calibrated (resp. uniformly calibrated, resp. consistent) at level $\varepsilon$ if $\phi_\varepsilon$ is calibrated (resp. uniformly calibrated, resp. consistent) wrt $l_{0/1,\varepsilon}$.*

Note that a first important sanity check to make is verify that $\phi_\varepsilon$ and $l_{0/1,\varepsilon}$ are indeed measurable and well defined. The arguments are not trivial since it uses advanced arguments from measure theory, but it is necessary to establish measurability before going further on. Proposition 2.1 states the measurability of $\phi_\varepsilon$ and $l_{0/1,\varepsilon}$. We prove this result in Appendix B.

**Proposition 2.1.** *Let $\phi : \mathbb{R} \times \mathcal{Y} \to \mathbb{R}$ be a measurable function and $\varepsilon \geq 0$. For every $f \in \mathcal{F}(\mathcal{X})$, $(x, y) \mapsto \phi_\varepsilon(x, y, f)$ and $(x, y) \mapsto l_{0/1,\varepsilon}(x, y, f)$ are universally measurable.*

Now that, we proved that the adversarial setting is properly defined, we can make a first observation: the calibration functions for $\phi$ and $\phi_\varepsilon$ are actually equal. This property might seem counter-intuitive at first sight as the adversarial risk is most of the time strictly larger than its standard counterpart. However, the calibration functions are only pointwise dependent, hence having the same prediction for any element of the ball $B_\varepsilon(x)$ suffices to reach the optimal calibration $\mathcal{C}^\star_\phi(x, \eta)$.

**Proposition 2.2.** *Let $\varepsilon > 0$. Let $\phi$ be a continuous classification margin loss. For all $x \in \mathcal{X}$ and $\eta \in [0, 1]$, we have*

$$\mathcal{C}^\star_{\phi_\varepsilon}(x, \eta) = \inf_{\alpha \in \mathbb{R}} \eta\phi(\alpha) + (1 - \eta)\phi(-\alpha) = \mathcal{C}^\star_\phi(x, \eta) \quad .$$

*The last equality also holds for the adversarial $0/1$ loss.*

The proof of this result is available in Appendix C.

## 3 Solving Adversarial Calibration

In this section, we study the calibration of adversarial margin losses with regard to the adversarial $0/1$ loss. We first provide necessary and sufficient conditions under which margin losses are adversarially calibrated. We then show that a wide range of surrogate losses that are calibrated in the standard setting are not calibrated in the adversarial setting. Finally we propose a class of losses that are calibrated in the adversarial setting, namely the *shifted odd losses*.

### 3.1 Necessary and Sufficient Conditions for Calibration

One of our main contributions is to find necessary and sufficient conditions for calibration in the adversarial setting. In a brief, we identify that for studying calibration it is central to understand the case where there might be indecision for classifiers (i.e. $\eta = 1/2$). Indeed, in this case, either labelling positively or negatively the input $x$ would lead the same loss for $x$. Next result provides a necessary condition for calibration.

**Theorem 3.1** (Necessary condition for Calibration). *Let $\phi$ be a continuous margin loss and $\varepsilon > 0$. If $\phi$ is adversarially calibrated at level $\varepsilon$, then $\phi$ is calibrated in the standard classification setting and $0 \notin \operatorname{argmin}_{\alpha \in \bar{\mathbb{R}}} \frac{1}{2}\phi(\alpha) + \frac{1}{2}\phi(-\alpha)$.*

We proof this theorem in Appendix D. While the condition of calibration in the standard classification setting seems natural, we need to understand why $0 \notin \operatorname{argmin}_{\alpha \in \bar{\mathbb{R}}} \frac{1}{2}\phi(\alpha) + \frac{1}{2}\phi(-\alpha)$. The intuition behind this result is that a sequence of functions simply converging towards $0$ in the ball of radius $\varepsilon$ around some $x$ can take positive and negative values thus leading to suboptimal $0/1$ adversarial risk. It turns out that, given an additional mild assumption, this condition is actually sufficient to ensure calibration.

**Theorem 3.2** (Sufficient condition for Calibration). *Let $\phi$ be a continuous margin loss and $\varepsilon > 0$. If $\phi$ is decreasing and strictly decreasing in a neighbourhood of $0$ and calibrated in the standard setting and $0 \notin \operatorname{argmin}_{\alpha \in \bar{\mathbb{R}}} \frac{1}{2}\phi(\alpha) + \frac{1}{2}\phi(-\alpha)$, then $\phi$ is adversarially uniformly calibrated at level $\varepsilon$.*

The proof of this theorem is available in Appendix E.

**Remark 2** (Decreasing hypothesis). *For the reciprocal, the additional assumption that $\phi$ is decreasing and strictly decreasing in a neighborhood of $0$ is not restrictive for usual losses. In Theorem 2.1, this assumption is stated as a necessary and sufficient condition for convex losses to be calibrated.*

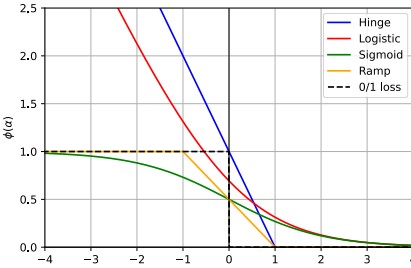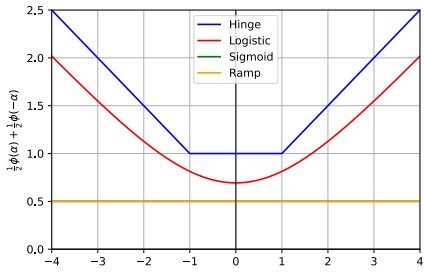

Figure 1: On the left, illustration of common standardly calibrated losses. On the right plot of their symmetrized version. Here we notice that $0 \in \mathrm{argmin}_\alpha \phi(\alpha) + \phi(-\alpha)$ for all these losses. Thus none of them are adversarially calibrated.

## 3.2 Negative results

Thanks to Theorem 3.1, we can present two notable corollaries invalidating the use of two important classes of surrogate losses in the standard setting. The first class of losses are convex margin losses. These losses are maybe the most widely used in modern day machine learning as they comprise the logistic loss or the margin loss that are the building block of most classification algorithms. We illustrate such losses in Figure 1.

**Corollary 3.1.** *Let $\varepsilon > 0$. Then no convex margin loss can be adversarially calibrated at level $\varepsilon$.*

A convex loss satisfies $\frac{1}{2}\phi(\alpha) + \frac{1}{2}\phi(-\alpha) \geq \phi(0)$, hence $0 \in \mathrm{argmin}_{\alpha \in \mathbb{R}} \phi(\alpha) + \phi(-\alpha)$. From Theorem 3.1, we deduce the result. Then, $\phi$ is not adversarially calibrated at level $\varepsilon$. This result seems counter-intuitive and highlights the difficulty of optimizing and understanding the adversarial risk. Since convex losses are not adversarially calibrated, one may hope to rely on famous non-convex losses such as sigmoid and ramp losses. But, unfortunately, such losses are not calibrated either as illusrtated in Figure 1.

**Corollary 3.2.** *Let $\varepsilon > 0$. Let $\lambda \in \mathbb{R}$ and $\psi$ be a lower-bounded odd function such that for all $\alpha \in \mathbb{R}$, $\psi > -\lambda$. We define $\psi$ as $\phi(\alpha) = \lambda + \psi(\alpha)$. Then $\phi$ is not adversarially calibrated at level $\varepsilon$.*

Indeed, $\frac{1}{2}\phi(\alpha) + \frac{1}{2}\phi(-\alpha) = \lambda$, so that $\mathrm{argmin}_{\alpha \in \mathbb{R}} \frac{1}{2}\phi(\alpha) + \frac{1}{2}\phi(-\alpha) = \mathbb{R}$. Thanks to Theorem 3.1, $\phi$ is not adversarially calibrated at level $\varepsilon$.

## 3.3 Positive results

Theorem 3.2 also gives sufficient conditions for $\phi$ to be adversarially calibrated. Leveraging this result, we devise a class of margin losses that are indeed calibrated in the adversarial settings. We call this class *shifted odd losses*, and we define it as follows.

**Definition 3.1** (Shifted odd losses). *We say that $\phi$ is a* shifted odd margin loss *if there exists $\lambda \geq 0$, $\tau > 0$, and a continuous lower bounded decreasing odd function $\psi$ that is strictly decreasing in a neighborhood of $0$ such that for all $\alpha \in \mathbb{R}$, $\psi(\alpha) \geq -\lambda$ and $\phi(\alpha) = \lambda + \psi(\alpha - \tau)$.*

The key difference between a standard odd margin loss and a shifted odd margin loss is the variations of the function $\alpha \mapsto \frac{1}{2}\phi(\alpha) + \frac{1}{2}\phi(-\alpha)$. The primary difference is that, in the standard case the optima of this function are located at $0$ while they are located in $-\infty$ and $+\infty$ in the adversarial setting. Let us give some examples of margin shifted odd losses below.

**Example** (Shifted odd losses). *For every $\varepsilon > 0$ and every $\tau > 0$, the shifted logistic loss, defined as follows, is adversarially calibrated at level $\varepsilon$: $\phi : \alpha \mapsto (1 + \exp(\alpha - \tau))^{-1}$ This loss is plotted on left in Figure 2. We also plotted on right in Figure 2 $\alpha \mapsto \frac{1}{2}\phi(\alpha) + \frac{1}{2}\phi(-\alpha)$ to justify that $0 \notin \mathrm{argmin}_{\alpha \in \bar{\mathbb{R}}} \frac{1}{2}\phi(\alpha) + \frac{1}{2}\phi(-\alpha)$. Also note that the shifted ramp loss also satisfies the same properties.*

A consequence of Theorem 3.2 is that shifted odd losses are adversarially calibrated, as demonstrated in Proposition 3.1 stated below.

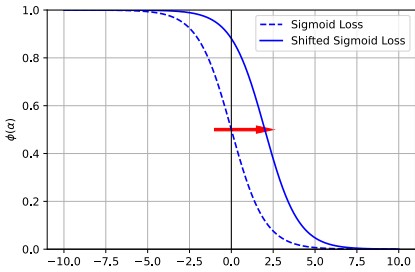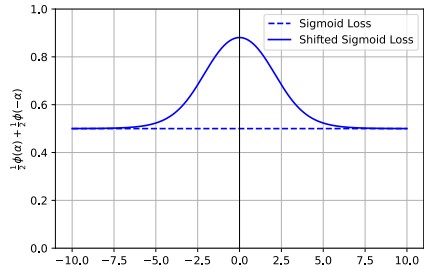

Figure 2: Illustration of a calibrated loss in the adversarial setting. The sigmoid loss satisfy the hypothesis for $\psi$. Its shifted version is then calibrated for adversarial classification.

**Proposition 3.1.** *Let $\phi$ be a shifted odd margin loss. For every $\varepsilon > 0$, $\phi$ is adversarially calibrated at level $\varepsilon$.*

We proof this proposition in Appendix F.

# 4 Towards Adversarial Consistency

We focus our study now on the problem of adversarial consistency. In a first part, taking inspiration from Long and Servedio [11], Awasthi et al. [1], we study the $\varepsilon$-realisable case, i.e. the case where the adversarial risk at level $\varepsilon$ equals zero. In a second part, we analyze the behavior of a candidate class of losses, namely the $0/1$-*like margin losses*.

## 4.1 The Realizable Case

The realizable case is important since there are no possible adversaries for the Bayes optimal classifier. Formally, this means that the adversarial risk equals $0$, as stated in the following definition.

**Definition 4.1** ($\varepsilon$-realisability)**.** *Let $\mathbb{P}$ be a Borel probability distribution on $\mathcal{X} \times \mathcal{Y}$ and $\varepsilon \geq 0$. We say that $\mathbb{P}$ is $\varepsilon$-realisable if $\mathcal{R}_{\varepsilon,\mathbb{P}}^{\star} = 0$.*

In the case of realizable probability distribution, calibrated (and consequently consistent) margin losses in the standard classification setting are also calibrated and consistent in the adversarial case.

**Proposition 4.1.** *Let $\varepsilon > 0$. Let $\mathbb{P}$ be an $\varepsilon$-realisable distribution and $\phi$ be a calibrated margin loss in the standard setting. Then $\phi$ is adversarially consistent at level $\varepsilon$.*

We give a proof of this result in Appendix G. The intuition behind this result is that if a probability distribution is $\varepsilon$-realisable, the marginal distributions are sufficiently separated, so that there are no possible adversarial attacks, each point in the $\varepsilon$-neighbourhood of the support of the distribution can be classified independently of each other.

## 4.2 Towards the General Case

In this section, we seek to pave the way towards proving the consistency of shifted odd losses. We will observe that their behavior is actually very similar to that of the $0/1$ loss, which makes them good candidates to be consistent losses. To this end, we first add an extra hypothesis to the odd shifted losses in order to simplify our technical analysis.

**Definition 4.2** ($0/1$-like margin losses)**.** *$\phi$ is a $0/1$-like margin loss if there exists $\lambda \geq 0$, $\tau \geq 0$, and a continuous lower bounded strictly decreasing odd function $\psi$ in a neighbourhood of $0$ such that for all $\alpha \in \mathbb{R}$, $\psi(\alpha) \geq -\lambda$ and $\phi(\alpha) = \lambda + \psi(\alpha - \tau)$ and*

$$\lim_{t \to -\infty} \phi(t) = 1 \text{ and } \lim_{t \to +\infty} \phi(t) = 0$$

Note here that the losses here are not necessarily shifted because $\tau$ might equal $0$, making this condition weaker. Consequently, we cannot hope that such losses are consistent neither calibrated,

but they might help in finding the path towards consistency. Note also that if $\phi$ is an odd or shifted odd loss, one can always find a rescaling of $\phi$ such that $\phi$ becomes a $0/1$-like margin loss. Note also that such a rescaling does neither change the notion of consistency and calibration for $\phi$ nor for its rescaled version.

Based on min-max arguments, we provide below some results better characterizing $0/1$-like margin loss functions in the adversarial setting. Let us first recall the notions of *midpoint property* and *adversarial distributions set* that will be useful from now on as well as an important existing result from Pydi and Jog [16].

**Definition 4.3.** *Let $(\mathcal{X}, d)$ be a proper Polish metric space. We say that $\mathcal{X}$ satisfy the* midpoint property *if for all $x_1, x_2 \in \mathcal{X}$ there exist $x \in \mathcal{X}$ such that $d(x, x_1) = d(x, x_2) = \frac{d(x_1, x_2)}{2}$.*

We recall also the set $\mathcal{A}_\varepsilon(\mathbb{P})$ of adversarial distributions introduced in [13].

**Definition 4.4.** *Let $\mathbb{P}$ be a Borel probability distribution and $\varepsilon > 0$. We define the set of* adversarial distributions $\mathcal{A}_\varepsilon(\mathbb{P})$ *as:*

$$\mathcal{A}_\varepsilon(\mathbb{P}) := \left\{ \mathbb{Q} \in \mathcal{M}_1^+(\mathcal{X} \times \mathcal{Y}) \mid \exists \gamma \in \mathcal{M}_1^+ \left( (\mathcal{X} \times \mathcal{Y})^2 \right), \right.$$
$$\left. d(x, x') \leq \varepsilon, \ y = y' \ \gamma \left( (x, y), (x', y') \right) \text{-a.s., } \Pi_{1\sharp}\gamma = \mathbb{P}, \ \Pi_{2\sharp}\gamma = \mathbb{Q} \right\}$$

*where $\Pi_i$ denotes the projection on the $i$-th component.*

Note that in this definition, $(x, y)$ denotes an input-label pair drawn from the original distribution $\mathbb{P}$ and $(x', y')$ denotes the input-label pair from the perturbed distribution $\mathbb{Q}$ (after attack)

**Theorem 4.1** (Pydi and Jog [16]). *Let $\mathcal{X}$ be a Polish space satisfying the midpoint property. Then strong duality holds:*

$$\mathcal{R}^\star_{\varepsilon, \mathbb{P}} = \inf_{f \in \mathcal{F}(\mathcal{X})} \sup_{\mathbb{Q} \in \mathcal{A}_\varepsilon(\mathbb{P})} \mathcal{R}_\mathbb{Q}(f) = \sup_{\mathbb{Q} \in \mathcal{A}_\varepsilon(\mathbb{P})} \inf_{f \in \mathcal{F}(\mathcal{X})} \mathcal{R}_\mathbb{Q}(f)$$

*Moreover the supremum of the right-hand term is attained.*

Note that in the original version of the theorem, Pydi and Jog [16] did not prove that the supremum is attained. We add a proof of this in Appendix H.

**Connections between $0/1$-like margin loss and $0/1$ loss: a min-max viewpoint.** Thanks the the above concepts, we can now present some results identifying the similarity and the differences between the $0/1$ loss and $0/1$-like margin losses. We first show that for a given fixed probability distribution $\mathbb{P}$, the adversarial optimal risk associated with a $0/1$-like margin loss and the $0/1$ loss are equal. The proof of this result is available in Appendix I

**Theorem 4.2.** *Let $\mathcal{X}$ be a Polish space satisfying the midpoint property. Let $\varepsilon \geq 0$, $\mathbb{P}$ be a Borel probability distribution over $\mathcal{X} \times \mathcal{Y}$, and $\phi$ be a $0/1$-like margin loss. Then, we have:*

$$\mathcal{R}^\star_{\phi_\varepsilon, \mathbb{P}} = \mathcal{R}^\star_{\varepsilon, \mathbb{P}}$$

In particular, we note that this property holds true for the standard risk. From this result, we can derive two interesting corollaries about $0/1$-like margin losses. First, strong duality holds for the risk associated with $\phi$.

**Corollary 4.1** (Strong duality for $\phi$). *Let us assume that $\mathcal{X}$ is a Polish space satisfying the midpoint property. Let $\varepsilon \geq 0$, $\mathbb{P}$ be a Borel probability distribution over $\mathcal{X} \times \mathcal{Y}$, and $\phi$ be a $0/1$-like margin loss. Then, we have:*

$$\inf_{f \in \mathcal{F}(\mathcal{X})} \sup_{\mathbb{Q} \in \mathcal{A}_\varepsilon(\mathbb{P})} \mathcal{R}_{\phi, \mathbb{Q}}(f) = \sup_{\mathbb{Q} \in \mathcal{A}_\varepsilon(\mathbb{P})} \inf_{f \in \mathcal{F}(\mathcal{X})} \mathcal{R}_{\phi, \mathbb{Q}}(f)$$

*Moreover the supremum is attained.*

Note that there is no reason that the infimum is attained. A second interesting corollary is the equality of the set of optimal attacks, i.e. distributions of $\mathcal{A}_\varepsilon(\mathbb{P})$ that maximize the dual problem: an optimal attack for the $0/1$ loss is also an optimal attack for a $0/1$-like margin, and vice versa.

**Corollary 4.2** (Optimal attacks). *Let assume that $\mathcal{X}$ be a Polish space satisfying the midpoint property. Let $\varepsilon \geq 0$ and $\mathbb{P}$ be a Borel probability distribution over $\mathcal{X} \times \mathcal{Y}$. Then, an optimal attack $\mathbb{Q}^\star$ of level $\varepsilon$ exists for both the $0/1$ loss and $\phi$. Moreover, for $\mathbb{Q} \in \mathcal{A}_\varepsilon(\mathbb{P})$. $\mathbb{Q}$ is an optimal attack for the loss $\phi$ if and only if it is an optimal attack for the $0/1$ loss.*

The proof of these two corollaries is available in Appendix J.

**A step towards consistency.** From the previous results, we are able to prove a first result toward the demonstration of consistency. This result is much weaker than consistency result, but it guarantees that if a sequence minimizes the adversarial risk, then it minimizes the risk for optimal attacks, i.e. in a game where the attacker plays before the classifier. The proof of this result is in Appendix K.

**Proposition 4.2.** *Let us assume that $\mathcal{X}$ be a Polish space satisfying the midpoint property. Let $\varepsilon \geq 0$ and $\mathbb{P}$ be a Borel probability distribution over $\mathcal{X} \times \mathcal{Y}$. Let $\mathbb{Q}^\star$ be an optimal attack of level $\varepsilon$. Let $(f_n)_{n \in \mathbb{N}}$ be a sequence of $\mathcal{F}(\mathcal{X})$ such that $\mathcal{R}_{\phi_\varepsilon, \mathbb{P}}(f_n) \to \mathcal{R}^\star_{\phi_\varepsilon, \mathbb{P}}$. Then $\mathcal{R}_{\mathbb{Q}^\star}(f_n) \to \mathcal{R}^\star_{\varepsilon, \mathbb{P}}$.*

We hope this result and its proof may lead to a full proof of consistency. This result is significantly weaker than consistency as stated in the following remark. In the proof of the previous results, we did not use the assumptions that losses are shifted. In our opinion, it is the key element that we miss and that we need to use to conclude on the consistency of this family of losses. The shift in the loss would force the classifier to goes to $\pm\infty$ on a $\varepsilon$ neighborhood support of the distribution of $\mathbb{P}$ and then the risk would equal the adversarial $0/1$-loss risk. However, we did not succeed in showing the appropriate result: we believe this question is complicated and is left as further work.

# 5 Related Work and Discussions

We now explain the differences between our approach and the one proposed by Bao et al. [4], Awasthi et al. [1, 3]. The two main differences are the choice of the $0/1$ loss and the studied notion of consistency and calibration.

**Alternative $0/1$ loss** An alternative $0/1$ loss would the following: $l_\leq(f(x), y) = \mathbf{1}_{yf(x)\leq 0}$. This loss penalizes indecision: i.e. predicting 0 would lead to a pointwise risk of 1 for $y = 1$ and $y = -1$ while the $0/1$ loss $l_{0/1}$ returns 1 for $y = 1$ and 0 for $y = -1$. This definition was used by Bao et al. [4], Awasthi et al. [1, 3] to prove their calibration and consistency results. While Bartlett et al. [6] was not explicit on the choice for the $0/1$ loss, Steinwart [17] explicitly mentions that the $0/1$ loss is not a margin loss. The use of this loss is not suited for studying consistency and leads to inaccurate results as shown in the following counterexample. On $\mathcal{X} = \mathbb{R}$, let $\mathbb{P}$ defined as $\mathbb{P} = \frac{1}{2}\left(\delta_{x=0,y=1} + \delta_{x=0,y=-1}\right)$ and $\phi : \mathbb{R} \to \mathbb{R}$ be a margin based loss. The $\phi$-risk minimization problem writes $\inf_\alpha \frac{1}{2}(\phi(\alpha) + \phi(-\alpha))$. For any convex functional $\phi$ the optimum is attained for $\alpha = 0$. $f_n : x \mapsto 0$ is a minimizing sequence for the $\phi$-risk. However $R_{l_\leq}(f_n) = 1$ for all $n$ and $R^*_{l_\leq} = \frac{1}{2}$. Then we deduce that no convex margin based loss is consistent wrt $l_\leq$. Consequently, the $0/1$ loss to be used in adversarial consistency needs to be $l_{0/1,\varepsilon}(x, y, f) = \sup_{x' \in B_\varepsilon(x)} \mathbf{1}_{y\text{sign}(f(x))\leq 0}$, otherwise the obtained results might be innacurate.

**$\mathcal{H}$-consistency and $\mathcal{H}$-calibration** Bao et al. [4], Awasthi et al. [1, 3] proposed to study $\mathcal{H}$-calibration and $\mathcal{H}$-consistency in the adversarial setting, i.e. calibration and consistency when minimizing sequences are in $\mathcal{H}$. However, even in the standard classification setting, the link between both notions in this extended setting is not clear at all since a pointwise minimization of the risk cannot be done. To our knowledge, there is only one research paper [11] that focuses on this notion in standard setting. They do it in the restricted case of realisability, i.e. when the standard optimal risk associated with the $0/1$ loss equals 0. We believe that studying $\mathcal{H}$-consistency and $\mathcal{H}$-calibration in the adversarial setting is a bit anticipated. For these reasons, we focus only on calibration and consistency on the space of measurable functions $\mathcal{F}(\mathcal{X})$. However, note that many of our results can be adapted to $\mathcal{H}$-calibration: we propose, in Appendix L, conditions on classes $\mathcal{H}$ so that Theorems 3.1 and 3.2 still holds.

**About the Adversarial Bayes Risk and Game Theory.** A recent trend of work has focused on analyzing the adversarial risk from multiple point of views. Bhagoji et al. [8] as well as Pydi and Jog [15, 16] showed that the adversarial optimal Bayes classifier can be written as optimal transport for a well chosen cost. Another line of work [14, 13, 16] have focused on a game theoretic approach for analyzing the adversarial risk having interest in the nature of equilibria between the classifier and the attacker. Recently, some researchers [2, 10] proved encouraging results on the existence of an optimal Bayes classifier in the adversarial setting under mild assumptions.

# 6 Conclusion and Perspectives

In this paper, we set some solid theoretical foundations for the study of adversarial consistency. We highlighted the importance of the definition of the $0/1$ loss, as well as the nuance between calibration and consistency that is specific to the adversarial setting. Furthermore, we solved the adversarial calibration problem, by giving a necessary and sufficient condition for decreasing, continuous margin losses to be adversarially calibrated. Since this is a necessary condition for consistency, an important consequence of this result is that no convex margin loss can be consistent. This rules out most of the commonly used surrogates, and spurs the need for new families of consistent, yet differentiable families of losses. We provide first insights into which losses may be consistent, by showing that translations of odd loss functions are calibrated.

Nonetheless, we believe that our work could have a concrete practical impact in the near future, especially when considering adversarial training [12]. When implementing this defense, we are essentially optimizing a surrogate adversarial loss. As our results show, these losses are not consistent when chosen with standard consistency theory, which can lead to a large optimality gap when using these techniques. We believe that using adversarially calibrated losses as the shifted odd losses could circumvent this problem, which would give important practical significance to our work.

## Acknowledgement

Rafael is partly supported by an Ecocloud postdoctoral fellowship.

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
