# Appendix

## A Equivalent definitions for calibration and consistency

**Consistency.** A loss $L_2$ is *consistent* wrt. a loss $L_1$ if and only if for every $\xi > 0$, there exists $\delta > 0$ such that for every $f \in \mathcal{F}(\mathcal{X})$,

$$\mathcal{R}_{L_2,\mathbb{P}}(f) - \mathcal{R}^{\star}_{L_2,\mathbb{P}} \leq \delta \implies \mathcal{R}_{L_1,\mathbb{P}}(f) - \mathcal{R}^{\star}_{L_1,\mathbb{P}} \leq \xi$$

**Calibration.** $L_2$ is *calibrated* with regards to $L_1$ if for all $\eta \in [0,1]$, $x \in \mathcal{X}$, for all $(f_n)_n \in \mathcal{F}(\mathcal{X})^{\mathbb{N}}$:

$$\mathcal{C}_{L_2}(x, \eta, f) - \mathcal{C}^{\star}_{L_2}(x, \eta) \xrightarrow[n \to \infty]{} 0 \implies \mathcal{C}_{L_1}(x, \eta, f) - \mathcal{C}^{\star}_{L_1}(x, \eta) \xrightarrow[n \to \infty]{} 0 \quad .$$

Also, $L_2$ is *uniformly calibrated* with regards to $L_1$ if for all $(f_n)_n \in \mathcal{F}(\mathcal{X})^{\mathbb{N}}$:

$$\sup_{\eta \in [0,1], x \in \mathcal{X}} \mathcal{C}_{L_2}(x, \eta, f) - \mathcal{C}^{\star}_{L_2}(x, \eta) \xrightarrow[n \to \infty]{} 0 \implies \sup_{\eta \in [0,1], x \in \mathcal{X}} \mathcal{C}_{L_1}(x, \eta, f) - \mathcal{C}^{\star}_{L_1}(x, \eta) \xrightarrow[n \to \infty]{} 0 \quad .$$

## B Proof of Proposition 2.1

**Proposition.** *Let $\phi : \mathbb{R} \times \mathcal{Y} \to \mathbb{R}$ be a measurable function and $\varepsilon \geq 0$. For every $f \in \mathcal{F}(\mathcal{X})$, $(x, y) \mapsto \phi_\varepsilon(x, y, f)$ and $(x, y) \mapsto l_{0/1,\varepsilon}(x, y, f)$ are universally measurable.*

*Proof.* Let $\phi : \mathbb{R} \to \mathbb{R}_+$ be a continuous function. We define $\phi_\varepsilon(x, y, f) = \sup_{x' \in B_\varepsilon(x)} \phi(y f(x))$.

We have :

$$\phi_\varepsilon(x, y, f) = \sup_{(x', y') \in \mathcal{X} \times \mathcal{Y}} \phi(y' f(x')) - \infty \times \mathbf{1}\{d(x', x) \geq \varepsilon \text{ or } y' \neq y\}$$

We have that

$$((x, y), (x', y')) \mapsto \phi(y' f(x')) - \infty \times \mathbf{1}\{d(x', x) \geq \varepsilon \text{ or } y' \neq y\}$$

defines a measurable, hence upper semi-analytic function. Using [7, Proposition 7.39, Corollary 7.42], we get that for all $f \in \mathcal{F}(\mathcal{X})$, $(x, y) \mapsto \phi_\varepsilon(x, y, f)$ is a universally measurable function. □

## C Proof of Proposition 2.2

**Proposition.** *Let $\varepsilon > 0$. Let $\phi$ be a continuous classification margin loss. For all $x \in \mathcal{X}$ and $\eta \in [0,1]$,*

$$\mathcal{C}^{\star}_{\phi_\varepsilon}(x, \eta) = \inf_{f \in \mathcal{F}(\mathcal{X})} \mathcal{C}_{\phi_\varepsilon}(x, \eta, f) = \inf_{\alpha \in \mathbb{R}} \eta \phi(\alpha) + (1 - \eta)\phi(-\alpha) = \mathcal{C}^{\star}_\phi(x, \eta) \quad .$$

*The last equality also holds for the $0/1$ loss.*

*Proof.* For any $f \in \mathcal{F}(\mathcal{X})$, we have:

$$\mathcal{C}_{\phi_\varepsilon}(x, \eta, f) = \eta \sup_{x' \in B_\varepsilon(x)} \phi(f(x')) + (1 - \eta) \sup_{x' \in B_\varepsilon(x)} \phi(-f(x'))$$

$$\geq \eta \phi(f(x)) + (1 - \eta)\phi(-f(x))$$

$$\geq \inf_{\alpha \in \mathbb{R}} \eta \phi(\alpha) + (1 - \eta)\phi(-\alpha) \quad .$$

Then we deduce that $\inf_{f \in \mathcal{F}(\mathcal{X})} \mathcal{C}_{\phi_\varepsilon}(x, \eta, f) \geq \inf_{\alpha \in \mathbb{R}} \eta \phi(\alpha) + (1 - \eta)\phi(-\alpha)$. On the other side, let $(\alpha_n)_n$ be a minimizing sequence such that $\eta \phi(\alpha_n) + (1 - \eta)\phi(-\alpha_n) \xrightarrow[n \to \infty]{} \inf_{\alpha \in \mathbb{R}} \eta \phi(\alpha) + (1 - \eta)\phi(-\alpha)$. We set $f_n : x \mapsto \alpha_n$ for all $x$. Then we have:

$$\inf_{f \in \mathcal{F}(\mathcal{X})} \mathcal{C}_{\phi_\varepsilon}(x, \eta, f) \leq \mathcal{C}_{\phi_\varepsilon}(x, \eta, f_n)$$

$$= \eta \phi(\alpha_n) + (1 - \eta)\phi(-\alpha_n) \xrightarrow[n \to \infty]{} \inf_{\alpha \in \mathbb{R}} \eta \phi(\alpha) + (1 - \eta)\phi(-\alpha).$$

Then we conclude that: $\mathcal{C}^{\star}_{\phi_\varepsilon}(x, \eta) = \inf_{\alpha \in \mathbb{R}} \eta \phi(\alpha) + (1 - \eta)\phi(-\alpha)$. □

# D    Proof of Theorem 3.1

**Theorem** (Necessary condition for Calibration). *Let $\phi$ be a continuous margin loss and $\varepsilon > 0$. If $\phi$ is adversarially calibrated at level $\varepsilon$, then $\phi$ is calibrated in the standard classification setting and $0 \notin \mathrm{argmin}_{\alpha \in \bar{\mathbb{R}}} \frac{1}{2}\phi(\alpha) + \frac{1}{2}\phi(-\alpha)$.*

*Proof.* Let us show that if $0 \in \mathrm{argmin}_{\alpha \in \mathbb{R}} \phi(\alpha) + \phi(-\alpha)$ then $\phi$ is not calibrated for the adversarial problem. For that, let $x \in \mathcal{X}$ and we fix $\eta = \frac{1}{2}$. For $n \geq 1$, we define $f_n(u) = \frac{1}{n}$ for $u \neq x$ and $-\frac{1}{n}$ for $u = x$. Since $|\mathcal{B}_\varepsilon(x)| \geq 2$, we have

$$\mathcal{C}_{\phi_\varepsilon}(x, \frac{1}{2}, f_n) = \max\left(\phi(\frac{1}{n}), \phi(-\frac{1}{n})\right) \xrightarrow[n \to \infty]{} \phi(0)$$

As, $\phi(0) = \inf_{\alpha \in \mathbb{R}} \frac{1}{2}(\phi(\alpha) + \phi(-\alpha))$, the above means that $(f_n)_n$ is a minimizing sequence for $\alpha \mapsto \frac{1}{2}(\phi(\alpha) + \phi(-\alpha))$. Then thanks to Proposition 2.2, $(f_n)_n$ is also a minimizing sequence for $f \mapsto \mathcal{C}_{\phi_\varepsilon}(x, \frac{1}{2}, f)$. However, for every integer $n$, we have $\mathcal{C}_\varepsilon(x, \frac{1}{2}, f_n) = 1 \neq \frac{1}{2}$. As $\inf_{f \in \mathcal{F}(\mathcal{X})} \mathcal{C}_\varepsilon(x, \frac{1}{2}, f) = \frac{1}{2}$, $\phi$ is not calibrated with regard to the $0/1$ loss in the adversarial setting at level $\varepsilon$. We also immediately notice that if $\phi$ is calibrated with regard to $0/1$ loss in the adversarial setting at level $\varepsilon$ then $\phi$ is calibrated in the standard setting. $\square$

# E    Proof of Theorem 3.2

**Theorem** (Sufficient condition for Calibration). *Let $\phi$ be a continuous margin loss and $\varepsilon > 0$. If $\phi$ is decreasing and strictly decreasing in a neighbourhood of $0$ and calibrated in the standard setting and $0 \notin \mathrm{argmin}_{\alpha \in \mathbb{R}} \frac{1}{2}\phi(\alpha) + \frac{1}{2}\phi(-\alpha)$, then $\phi$ is adversarially uniformly calibrated at level $\varepsilon$.*

*Proof.* Let $\xi \in (0, \frac{1}{2})$. Thanks to Theorem 2.1, $\phi$ is uniformly calibrated in the standard setting, then there exists $\delta > 0$, such that for all $x \in \mathcal{X}, \eta \in [0, 1], f \in \mathcal{F}(\mathcal{X})$:

$$\mathcal{C}_\phi(x, \eta, f) - \mathcal{C}_\phi^\star(x, \eta) \leq \delta \implies \mathcal{C}(x, \eta, f) - \mathcal{C}^\star(x, \eta) \leq \xi.$$

**Case $\eta \neq \frac{1}{2}$:** Let $x \in \mathcal{X}$ and $f \in \mathcal{F}(\mathcal{X})$ such that:

$$\mathcal{C}_{\phi_\varepsilon}(x, \eta, f) - \mathcal{C}_{\phi_\varepsilon}^\star(x, \eta) = \sup_{u, v \in B_\varepsilon(x)} \eta\phi(f(u)) + (1 - \eta)\phi(-f(v)) - \mathcal{C}_{\phi_\varepsilon}^\star(x, \eta) \leq \delta$$

We recall thanks to Proposition 2.2 that for every $u, v \in \mathcal{X}$,

$$\mathcal{C}_{\phi_\varepsilon}^\star(u, \eta) = \mathcal{C}_\phi^\star(v, \eta) = \inf_{\alpha \in \mathbb{R}} \eta\phi(\alpha) + (1 - \eta)\phi(-\alpha) \quad .$$

Then in particular, for all $x' \in B_\varepsilon(x)$, we have:

$$\mathcal{C}_\phi(x', \eta, f) - \mathcal{C}_\phi^\star(x', \eta) \leq \sup_{u, v \in B_\varepsilon(x)} \eta\phi(f(u)) + (1 - \eta)\phi(-f(v)) - \mathcal{C}_{\phi_\varepsilon}^\star(x, \eta)$$

$$\leq \delta \quad .$$

Then since $\phi$ is calibrated for standard classification, for all $x' \in B_\varepsilon(x), \mathcal{C}(x', \eta, f) - \mathcal{C}^\star(x', \eta) \leq \xi$. Since, $\xi < \frac{1}{2}$, we have $\mathcal{C}(x', \eta, f) = \mathcal{C}^\star(x', \eta)$ and then for all $x' \in B_\varepsilon(x)$, $f(x') < 0$ if $\eta < 1/2$ or $f(x') \geq 0$ if $\eta > 1/2$. We then deduce that

$$\mathcal{C}_\varepsilon(x, \eta, f) = \eta \sup_{x' \in B_\varepsilon(x)} \mathbf{1}_{f(x') \leq 0} + (1 - \eta) \sup_{x' \in B_\varepsilon(x)} \mathbf{1}_{f(x') > 0}$$

$$= \min(\eta, 1 - \eta) = \mathcal{C}_\varepsilon^\star(x, \eta)$$

Then we deduce, $\mathcal{C}_\varepsilon(x, \eta, f) - \mathcal{C}_\varepsilon^\star(x, \eta) \leq \xi$.

**Case $\eta = \frac{1}{2}$:** This shows us that calibration problems will only arise when $\eta = \frac{1}{2}$, i.e. on points where the Bayes classifier is indecise. For this case, we will reason by contradiction: we can construct

a sequence of points $\alpha_n$ and $\beta_n$, whose risks converge to the same optimal value, while one sequence remains close to some positive value, and the other to some negative value. Assume that for all $n$, there exist $f_n \in \mathcal{F}(\mathcal{X})$ and $x_n \in \mathcal{X}$ such that

$$\mathcal{C}_{\phi_\varepsilon}\left(x_n, \frac{1}{2}, f_n\right) - \mathcal{C}_{\phi_\varepsilon}^\star\left(x_n, \frac{1}{2}\right) \leq \frac{1}{n}$$

and there exists $u_n, v_n \in B_\varepsilon(x_n)$, such that

$$f_n(u_n) f_n(v_n) \leq 0$$

Let us denote $\alpha_n = f_n(u_n)$ and $\beta_n = f_n(v_n)$. Moreover, we have thanks to Proposition 2.2:

$$0 \leq \frac{1}{2}\phi(\alpha_n) + \frac{1}{2}\phi(-\alpha_n) - \inf_{u \in \mathbb{R}}\left[\frac{1}{2}\phi(u) + \frac{1}{2}\phi(u)\right] \leq \mathcal{C}_{\phi_\varepsilon}\left(x, \frac{1}{2}, f_n\right) - \mathcal{C}_{\phi_\varepsilon}^\star\left(x, \frac{1}{2}\right)$$

$$\leq \frac{1}{n}$$

Then we deduce that $(\alpha_n)_n$ is a minimizing sequence for $u \mapsto \frac{1}{2}\phi(u) + \frac{1}{2}\phi(-u)$ and similarly $(\beta_n)_n$ is also a minimizing sequence for $u \mapsto \frac{1}{2}\phi(u) + \frac{1}{2}\phi(-u)$. Now note that there always exist $\alpha, \beta \in \bar{\mathbb{R}}$ such that, up to an extraction of a subsequence, we have $\alpha_n \xrightarrow[n \to \infty]{} \alpha$ and $\beta_n \xrightarrow[n \to \infty]{} \beta$. Furthermore by continuity of $\phi$ and since $0 \notin \arg\min \phi(u) + \phi(-u)$, $\alpha \neq 0$ and $\beta \neq 0$. Without loss of generality one can assume that $\alpha < 0 < \beta$, then for $n$ sufficiently large, $\alpha_n < 0 < \beta_n$. Moreover we have

$$0 \leq \frac{1}{2}\max\left(\phi(\alpha_n), \phi(\beta_n)\right) + \frac{1}{2}\max\left(\phi(-\alpha_n), \phi(-\beta_n)\right) - \mathcal{C}_{\phi_\varepsilon}^\star\left(x, \frac{1}{2}\right)$$

$$\leq \mathcal{C}_{\phi_\varepsilon}\left(x, \frac{1}{2}, f_n\right) - \mathcal{C}_{\phi_\varepsilon}^\star\left(x, \frac{1}{2}\right) \leq \frac{1}{n}$$

so that we deduce:

$$\frac{1}{2}\max\left(\phi(\alpha_n), \phi(\beta_n)\right) + \frac{1}{2}\max\left(\phi(-\alpha_n), \phi(-\beta_n)\right) \longrightarrow \inf_{u \in \mathbb{R}}\left[\frac{1}{2}\phi(u) + \frac{1}{2}\phi(u)\right] \qquad (2)$$

Since, for $n$ sufficiently large, $\alpha_n < 0 < \beta_n$ and $\phi$ is decreasing and strictly decreasing in a neighbourhood of $0$, we have that:

$$\max\left(\phi(\alpha_n), \phi(\beta_n)\right) = \phi(\alpha_n)$$

and

$$\max\left(\phi(-\alpha_n), \phi(-\beta_n)\right) = \phi(-\beta_n).$$

Moreover, there exists $\lambda > 0$ such that for $n$ sufficiently large $\phi(\alpha_n) - \phi(\beta_n) \geq \lambda$. Then for $n$ sufficiently large:

$$\frac{1}{2}\max\left(\phi(\alpha_n), \phi(\beta_n)\right) + \frac{1}{2}\max\left(\phi(-\alpha_n), \phi(-\beta_n)\right)$$

$$= \frac{1}{2}\phi(\alpha_n) + \frac{1}{2}\phi(-\beta_n)$$

$$= \frac{1}{2}\left(\phi(\alpha_n) - \phi(\beta_n)\right) + \frac{1}{2}\phi(-\beta_n) + \frac{1}{2} + \phi(\beta_n)$$

$$\geq \frac{1}{2}\lambda + \inf_{u \in \mathbb{R}}\left[\frac{1}{2}\phi(u) + \frac{1}{2}\phi(u)\right]$$

which leads to a contradiction with Equation 2. Then there exists a non zero integer $n_0$ such that for all $f \in \mathcal{F}(\mathcal{X})$, $x \in \mathcal{X}$

$$\mathcal{C}_{\phi_\varepsilon}\left(x, \frac{1}{2}, f\right) - \mathcal{C}_{\phi_\varepsilon}^\star\left(x, \frac{1}{2}\right) \leq \frac{1}{n_0} \implies \forall u, v \in B_\varepsilon(x), \ f(u) \times f(v) > 0.$$

The right-hand term is equivalent to: for all $u \in B_\varepsilon(x)$, $f(u) > 0$ or for all $u \in B_\varepsilon(x)$, $f(u) < 0$. Then $\mathcal{C}_\varepsilon(x, \eta, f) = \frac{1}{2}$ and then $\mathcal{C}_\varepsilon(x, \eta, f) = \mathcal{C}_\varepsilon^\star(x, \eta)$

Putting all that together, for all $x \in \mathcal{X}$, $\eta \in [0, 1]$, $f \in \mathcal{F}(\mathcal{X})$:

$$\mathcal{C}_{\phi_\varepsilon}(x, \eta, f) - \mathcal{C}_{\phi_\varepsilon}^\star(x, \eta) \leq \min\left(\delta, \frac{1}{n_0}\right) \implies \mathcal{C}_\varepsilon(x, \eta, f) - \mathcal{C}_\varepsilon^\star(x, \eta) \leq \xi.$$

Then $\phi$ is adversarially uniformly calibrated at level $\varepsilon$ $\qquad\qquad \square$

# F    Proof of Proposition 3.1

**Proposition.** *Let $\phi$ be a shifted odd margin loss. For every $\varepsilon > 0$, $\phi$ is adversarially calibrated at level $\varepsilon$.*

*Proof.* Let $\lambda > 0$, $\tau > 0$ and $\phi$ be a strictly decreasing odd function such that $\tilde{\phi}$ defined as $\tilde{\phi}(\alpha) = \lambda + \phi(\alpha - \tau)$ is non-negative.

**Proving that** $0 \notin \operatorname{argmin}_{t \in \bar{\mathbb{R}}} \frac{1}{2}\tilde{\phi}(t) + \frac{1}{2}\tilde{\phi}(-t)$**.**   $\phi$ is clearly strictly decreasing and non-negative then it admits a limit $l := -\lim_{t \to +\infty} \tilde{\phi}(t) \geq 0$. Then we have:

$$\lim_{t \to +\infty} \tilde{\phi}(t) = \lambda + l \quad \text{and} \quad \lim_{t \to -\infty} \tilde{\phi}(t) = \lambda - l$$

Consequently we have:

$$\lim_{t \to \infty} \frac{1}{2}\tilde{\phi}(t) + \frac{1}{2}\tilde{\phi}(-t) = \lambda$$

On the other side $\tilde{\phi}(0) = \lambda + \phi(-\tau) > \lambda + \phi(0) = \lambda$ since $\tau > 0$ and $\phi$ is strictly decreasing. Then $0 \notin \operatorname{argmin}_{t \in \bar{\mathbb{R}}} \frac{1}{2}\tilde{\phi}(t) + \frac{1}{2}\tilde{\phi}(-t)$.

**Proving that** $\tilde{\phi}$ **is calibrated for standard classification.**   Let $\xi > 0$, $\eta \in [0, 1]$, $x \in \mathcal{X}$. If $\eta = \frac{1}{2}$, then for all $f \in \mathcal{F}(\mathcal{X})$, $\mathcal{C}(x, \frac{1}{2}, f) = \mathcal{C}^\star(x, \frac{1}{2}) = \frac{1}{2}$. Let us now assume that $\eta \neq \frac{1}{2}$, we have for all $f \in \mathcal{F}(\mathcal{X})$:

$$\mathcal{C}_{\tilde{\phi}}(x, \eta, f) = \lambda + \eta\phi(f(x) - \tau) + (1 - \eta)\phi(-f(x) - \tau)$$

$$= \lambda + \left(\eta - \frac{1}{2}\right)\left(\phi(f(x) - \tau) - \phi(-f(x) - \tau)\right)$$

$$+ \frac{1}{2}\left(\phi(f(x) - \tau) + \phi(-f(x) - \tau)\right)$$

Let us show that $\operatorname{argmin}_{t \in \bar{\mathbb{R}}} \frac{1}{2}\tilde{\phi}(t) + \frac{1}{2}\tilde{\phi}(-t) = \{-\infty, +\infty\}$. We have for all $t$:

$$\frac{1}{2}\tilde{\phi}(t) + \frac{1}{2}\tilde{\phi}(-t) = \lambda + \frac{1}{2}\left(\phi(t - \tau) + \phi(-t - \tau)\right)$$

$$= \lambda + \frac{1}{2}\left(\phi(t - \tau) - \phi(t + \tau)\right) > \lambda$$

since $t - \tau < t + \tau$ and $\phi$ is strictly decreasing. Hence by continuity of $\phi$ the optimum are attained when $t \to \infty$ or $t \to -\infty$. Then $\operatorname{argmin}_{t \in \bar{\mathbb{R}}} \frac{1}{2}\tilde{\phi}(t) + \frac{1}{2}\tilde{\phi}(-t) = \{-\infty, +\infty\}$.

Without loss of generality, let $\eta > 1/2$, then

$$t \mapsto \left(\eta - \frac{1}{2}\right)\left(\phi(t - \tau) - \phi(-t - \tau)\right)$$

is strictly decreasing and $\operatorname{argmin}_{t \in \bar{\mathbb{R}}} \frac{1}{2}\left(\phi(t - \tau) + \phi(-t - \tau)\right) = \{-\infty, +\infty\}$, then we have

$$\operatorname*{argmin}_{t \in \bar{\mathbb{R}}} \lambda + \left(\eta - \frac{1}{2}\right)(t - \tau) - \phi(-t - \tau)) + \frac{1}{2}\left(\phi(t - \tau) + \phi(-t - \tau)\right) = \{+\infty\} \quad .$$

By continuity of $\phi$, we deduce that for $\delta > 0$ sufficiently small:

$$\mathcal{C}_{\tilde{\phi}}(x, \eta, f) - \mathcal{C}^\star_{\tilde{\phi}}(x, \eta) \leq \delta \implies f(x) > 0$$

The same reasoning holds for $\eta < \frac{1}{2}$. Then we deduce that $\tilde{\phi}$ is calibrated for standard classification.

Finally, we obtain that $\tilde{\phi}$ is calibrated for adversarial classification for every $\varepsilon > 0$.    $\square$

# G   Proof of Proposition 4.1

**Proposition.** *Let $\varepsilon > 0$. Let $\mathbb{P}$ be an $\varepsilon$-realisable distribution and $\phi$ be a calibrated margin loss in the standard setting. Then $\phi$ is adversarially consistent at level $\varepsilon$.*

To formally prove this result, we need a preliminary lemma.

**Lemma G.1.** *Let $\mathbb{P}$ be an $\varepsilon$-realisable distribution and $\phi$ be a calibrated margin loss in the standard setting. Then $\mathcal{R}^{\star}_{\phi_\varepsilon,\mathbb{P}} = \inf_{\alpha \in \mathbb{R}} \phi(\alpha)$.*

*Proof.* Let $a \in \mathbb{R}$ be such that $\phi(a) - \inf_{\alpha \in \mathbb{R}} \phi(\alpha) \leq \xi$. $\mathbb{P}$ being $\varepsilon$-realisable, there exists a measurable function $f$ such that:

$$\mathcal{R}_{\varepsilon,\mathbb{P}}(f) = \mathbb{E}_{\mathbb{P}}\left[\sup_{x' \in B_\varepsilon(x)} \mathbf{1}_{y\,\mathrm{sign}(f(x))\leq 0}\right] = \mathbb{P}\left[\exists x' \in B_\varepsilon(x), \mathrm{sign}(f(x')) \neq y\right]$$

$$\leq \xi' := \frac{\xi}{\max(1, \phi(-a))}.$$

Denoting $p = \mathbb{P}(y = 1)$, $\mathbb{P}_1 = \mathbb{P}[\cdot|y = 1]$ and $\mathbb{P}_{-1} = \mathbb{P}[\cdot|y = -1]$, we have:

$$p \times \mathbb{P}_1\left[\exists x' \in B_\varepsilon(x), f(x') < 0\right] \leq \xi'$$

and

$$(1-p) \times \mathbb{P}_{-1}\left[\exists x' \in B_\varepsilon(x), f(x') \geq 0\right] \leq \xi' \quad .$$

Let us now define $g$ as:

$$g(x) = \begin{cases} a \text{ if } f(x) \geq 0 \\ -a \text{ if } f(x) < 0 \end{cases}$$

We have:

$$\mathcal{R}_{\phi_\varepsilon,\mathbb{P}}(g) = \mathbb{E}_{\mathbb{P}}\left[\sup_{x' \in B_\varepsilon(x)} \phi(yg(x))\right]$$

$$= p \times \mathbb{E}_{\mathbb{P}_1}\left[\sup_{x' \in B_\varepsilon(x)} \phi(g(x))\right] + (1-p) \times \mathbb{E}_{\mathbb{P}_{-1}}\left[\sup_{x' \in B_\varepsilon(x)} \phi(-g(x))\right]$$

We have:

$$p \times \mathbb{E}_{\mathbb{P}_1}\left[\sup_{x' \in B_\varepsilon(x)} \phi(g(x))\right]$$

$$\leq p \times \mathbb{E}_{\mathbb{P}_1}\left[\sup_{x' \in B_\varepsilon(x)} \phi(g(x))\mathbf{1}_{f(x')<0}\right] + p \times \mathbb{E}_{\mathbb{P}_1}\left[\sup_{x' \in B_\varepsilon(x)} \phi(g(x))\mathbf{1}_{f(x')\geq 0}\right]$$

$$= \phi(-a) \times p \times \mathbb{P}_1\left[\exists x' \in B_\varepsilon(x), f(x') < 0\right]$$
$$+ \phi(a) \times p \times (1 - \mathbb{P}_1\left[\exists x' \in B_\varepsilon(x), f(x') < 0\right])$$
$$\leq \phi(-a)\xi' + p \times \phi(a)$$
$$\leq p \times \inf_{\alpha \in \mathbb{R}} \phi(\alpha) + 2\xi$$

Similarly, we have:

$$(1-p) \times \mathbb{E}_{\mathbb{P}_{-1}}\left[\sup_{x' \in B_\varepsilon(x)} \phi(-g(x))\right] \leq (1-p) \times \inf_{\alpha \in \mathbb{R}} \phi(\alpha) + 2\xi$$

We get: $\mathcal{R}_{\phi_\varepsilon,\mathbb{P}}(g) \leq \inf_{\alpha \in \mathbb{R}} \phi(\alpha) + 4\xi$ and, hence $\mathcal{R}^{\star}_{\phi_\varepsilon,\mathbb{P}} = \inf_{\alpha \in \mathbb{R}} \phi(\alpha)$. $\qquad\qquad\square$

We are now ready to prove the result of consistency in the realizable case.

*Proof.* Let $0 < \xi < 1$. Thanks to Theorem 2.1, $\phi$ is uniformly calibrated for standard classification, then, there exists $\delta > 0$ such that for all $f \in \mathcal{F}(\mathcal{X})$ and for all $x$:

$$\phi(yf(x)) - \inf_{\alpha \in \mathbb{R}} \phi(\alpha) \leq \delta \implies \mathbf{1}_{y\text{sign}f(x) \leq 0} = 0$$

Let now $f \in \mathcal{F}(\mathcal{X})$ be such that $\mathcal{R}_{\phi_\varepsilon, \mathbb{P}}(f) \leq \mathcal{R}_{\phi_\varepsilon, \mathbb{P}}^\star + \delta\xi$. Thanks to Lemma G.1, we have:

$$\mathcal{R}_{\phi_\varepsilon, \mathbb{P}}(f) - \mathcal{R}_{\phi_\varepsilon, \mathbb{P}}^\star = \mathbb{E}_{\mathbb{P}}\left[\sup_{x' \in B_\varepsilon(x)} \phi(yf(x)) - \inf_{\alpha \in \mathbb{R}} \phi(\alpha)\right] \leq \delta\xi$$

Then by Markov inequality:

$$\mathbb{P}\left[\sup_{x' \in B_\varepsilon(x)} \phi(yf(x)) - \inf_{\alpha \in \mathbb{R}} \phi(\alpha) \geq \delta\right] \leq \frac{\mathbb{E}_{\mathbb{P}}\left[\sup_{x' \in B_\varepsilon(x)} \phi(yf(x)) - \inf_{\alpha \in \mathbb{R}} \phi(\alpha)\right]}{\delta}$$
$$\leq \xi$$

So we have $\mathbb{P}\left[\forall x' \in B_\varepsilon(x), \phi(yf(x)) - \inf_{\alpha \in \mathbb{R}} \phi(\alpha) \leq \delta\right] \geq 1 - \xi$ and then

$$\mathbb{P}\left[\forall x' \in B_\varepsilon(x), \mathbf{1}_{y\text{sign}(f(x)) \leq 0} = 0\right] \geq 1 - \xi \quad .$$

Since $\mathbb{P}$ is $\varepsilon$-realisable, we have $\mathcal{R}_{\varepsilon, \mathbb{P}}^\star = 0$ and:

$$\mathcal{R}_{\varepsilon, \mathbb{P}}(f) - \mathcal{R}_{\varepsilon, \mathbb{P}}^\star = \mathcal{R}_{\varepsilon, \mathbb{P}}(f) = \mathbb{P}\left[\exists x' \in B_\varepsilon(x), \text{sign}(f(x')) \neq y\right] \leq \xi$$

which concludes the proof. $\qquad\qquad\square$

# H    Proof of the existence of a maximum in Theorem 4.1

**Theorem** (Pydi and Jog [16]). *Let $\mathcal{X}$ be a Polish space satisfying the midpoint property. Then strong duality holds:*

$$\mathcal{R}_\varepsilon^\star(\mathbb{P}) = \inf_{f \in \mathcal{F}(\mathcal{X})} \sup_{\mathbb{Q} \in \mathcal{A}_\varepsilon(\mathbb{P})} \mathcal{R}_{\mathbb{Q}}(f) = \sup_{\mathbb{Q} \in \mathcal{A}_\varepsilon(\mathbb{P})} \inf_{f \in \mathcal{F}(\mathcal{X})} \mathcal{R}_{\mathbb{Q}}(f)$$

*Moreover the supremum of the right-hand term is attained.*

We give a proof for the existence of the supremum.

*Proof.* To prove that, note that for every Borel probability distribution $\mathbb{Q}$ over $\mathcal{X} \times \mathcal{Y}$,

$$\inf_{f \in \mathcal{F}(\mathcal{X})} \mathcal{R}_{\mathbb{Q}}(f) = (1 - q) + \inf_{f \in \mathcal{C}(\mathcal{X}), \, 0 \leq f \leq 1} \int f d(q\mathbb{Q}_1 + (q-1)\mathbb{Q}_{-1})$$

where $\mathcal{C}(\mathcal{X})$ denotes the space of continuous functions on $\mathcal{X}$, $q = \mathbb{Q}[y = 1]$ and $\mathbb{Q}_i = \mathbb{Q}[\cdot \mid y = i]$. When $f$ is continuous and bounded, the function:

$$\mu \in \mathcal{M}(\mathcal{X}) \mapsto \int f d\mu$$

is continuous for the weak topology of measures, then:

$$\mu \in \mathcal{M}(\mathcal{X}) \mapsto \inf_{f \in \mathcal{C}(\mathcal{X}), \, 0 \leq f \leq 1} \int f d\mu$$

is upper semi continuous for the weak topology of measures, as it is the infinum of continuous functions. Then using the compacity of $\mathcal{A}_\varepsilon(\mathbb{P})$, we deduce that the supremum is attained. $\qquad\square$

# I Proofs of Theorem 4.2

**Theorem.** *Let $\mathcal{X}$ be a Polish space satisfying the midpoint property. Let $\varepsilon \geq 0$, $\mathbb{P}$ be a Borel probability distribution over $\mathcal{X} \times \mathcal{Y}$, and $\phi$ be a $0/1$-like margin loss. Then, we have:*

$$\mathcal{R}^\star_{\phi_\varepsilon, \mathbb{P}} = \mathcal{R}^\star_{\varepsilon, \mathbb{P}}$$

To prove this result, we need the following lemma.

**Lemma I.1.** *Let $\mathbb{Q}$ be a Borel probability distribution over $\mathcal{X} \times \mathcal{Y}$ and $\phi$ be a $0/1$-like margin loss, then: $\mathcal{R}^\star_{\phi, \mathbb{Q}} = \mathcal{R}^\star_{\mathbb{Q}}$.*

*Proof.* Bartlett et al. [6], Steinwart [17] proved that: for every margin losses $\phi$,

$$\mathcal{R}^\star_{\phi, \mathbb{Q}} = \inf_{f \in \mathcal{F}(\mathcal{X})} \mathbb{E}_{(x,y) \sim \mathbb{Q}} \left[ \phi(yf(x)) \right]$$

$$= \mathbb{E}_{x \sim \mathbb{Q}_x} \left[ \inf_{\alpha \in \mathbb{R}} \left[ \mathbb{Q}(y = 1|x)\phi(\alpha) + (1 - \mathbb{Q}(y = -1|x))\phi(-\alpha) \right] \right]$$

$$= \mathbb{E}_{x \sim \mathbb{Q}_x} \left[ \mathcal{C}^\star_\phi(\mathbb{Q}(y = 1|x), x) \right]$$

We also have $\mathcal{R}^\star_{\mathbb{Q}} = \mathbb{E}_{x \sim \mathbb{Q}_x} \left[ \mathcal{C}^\star(\mathbb{Q}(y = 1|x), x) \right]$. Moreover, if $\phi$ is a $0/1$-like margin loss, one can prove easily that for every $x \in \mathcal{X}$ and $\eta \in [0, 1]$, $\mathcal{C}^\star_\phi(\eta, x) = \min(\eta, 1 - \eta) = \mathcal{C}^\star(\eta, x)$. We can then conclude that $\mathcal{R}^\star_{\phi, \mathbb{Q}} = \mathcal{R}^\star_{\mathbb{Q}}$. $\qquad\square$

We can now prove Theorem 4.2.

*Proof.* Let $\xi > 0$ and $\mathbb{P}$ be a Borel probability distribution over $\mathcal{X} \times \mathcal{Y}$. Let $f$ such that $\mathcal{R}_{\varepsilon, \mathbb{P}}(f) \leq \mathcal{R}^\star_{\varepsilon, \mathbb{P}} + \xi$. Let $a > 0$ such that $\phi(a) \leq \xi$ and $\phi(-a) \geq 1 - \xi$. We define $g$ as:

$$g(x) = \begin{cases} a & \text{if } f(x) \geq 0 \\ -a & \text{if } f(x) < 0 \end{cases}$$

We have $\phi(yg(x)) = \phi(a)\mathbf{1}_{y\,sign(f(x))>0} + \phi(-a)\mathbf{1}_{y\,sign(f(x))\leq 0}$. Then

$$\mathcal{R}_{\phi_\varepsilon, \mathbb{P}}(g) = \mathbb{E}_\mathbb{P} \left[ \sup_{x' \in B_\varepsilon(x)} \phi(yg(x)) \right]$$

$$= \mathbb{E}_\mathbb{P} \left[ \sup_{x' \in B_\varepsilon(x)} \phi(-a)\mathbf{1}_{y\,sign(f(x'))\leq 0} + \phi(a)\mathbf{1}_{y\,sign(f(x'))>0} \right]$$

$$\leq \mathbb{E}_\mathbb{P} \left[ \sup_{x' \in B_\varepsilon(x)} \mathbf{1}_{y\,sign(f(x'))\leq 0} \right] + \phi(a)$$

$$\leq \mathcal{R}^\star_{\varepsilon, \mathbb{P}} + 2\xi \quad .$$

Then we have $\mathcal{R}^\star_{\phi_\varepsilon, \mathbb{P}} \leq \mathcal{R}^\star_{\varepsilon, \mathbb{P}}$. On the other side, we have:

$$\mathcal{R}^\star_{\phi_\varepsilon, \mathbb{P}} \geq \sup_{\mathbb{Q} \in \mathcal{A}_\varepsilon(\mathbb{P})} \inf_{f \in \mathcal{F}(\mathcal{X})} \mathcal{R}_{\phi, \mathbb{Q}}(f) = \sup_{\mathbb{Q} \in \mathcal{A}_\varepsilon(\mathbb{P})} \mathcal{R}^\star_{\phi, \mathbb{Q}}$$

$$= \sup_{\mathbb{Q} \in \mathcal{A}_\varepsilon(\mathbb{P})} \mathcal{R}^\star_{\mathbb{Q}} \text{ from Lemma I.1}$$

$$= \sup_{\mathbb{Q} \in \mathcal{A}_\varepsilon(\mathbb{P})} \inf_{f \in \mathcal{F}(\mathcal{X})} \mathcal{R}_{\mathbb{Q}}(f)$$

$$= \inf_{f \in \mathcal{F}(\mathcal{X})} \sup_{\mathbb{Q} \in \mathcal{A}_\varepsilon(\mathbb{P})} \mathcal{R}_{\mathbb{Q}}(f) = \mathcal{R}^\star_{\varepsilon, \mathbb{P}}$$

The last step is a consequence of Theorem 4.1. Then finally we get that $\mathcal{R}^\star_{\phi_\varepsilon, \mathbb{P}} = \mathcal{R}^\star_{\varepsilon, \mathbb{P}}$.

$\square$

# J  Proofs of Corollaries 4.1 and 4.2

**Corollary** (Strong duality for $\phi$). *Let us assume that $\mathcal{X}$ is a Polish space satisfying the midpoint property. Let $\varepsilon \geq 0$, $\mathbb{P}$ be a Borel probability distribution over $\mathcal{X} \times \mathcal{Y}$, and $\phi$ be a 0/1-like margin loss. Then, we have:*

$$\inf_{f \in \mathcal{F}(\mathcal{X})} \sup_{\mathbb{Q} \in \mathcal{A}_\varepsilon(\mathbb{P})} \mathcal{R}_{\phi, \mathbb{Q}}(f) = \sup_{\mathbb{Q} \in \mathcal{A}_\varepsilon(\mathbb{P})} \inf_{f \in \mathcal{F}(\mathcal{X})} \mathcal{R}_{\phi, \mathbb{Q}}(f)$$

*Moreover the supremum is attained.*

**Corollary** (Optimal attacks). *Let assume that $\mathcal{X}$ be a Polish space satisfying the midpoint property. Let $\varepsilon \geq 0$ and $\mathbb{P}$ be a Borel probability distribution over $\mathcal{X} \times \mathcal{Y}$. Then, an optimal attack $\mathbb{Q}^\star$ of level $\varepsilon$ exists for both the 0/1 loss and $\phi$. Moreover, for $\mathbb{Q} \in \mathcal{A}_\varepsilon(\mathbb{P})$. $\mathbb{Q}$ is an optimal attack for the loss $\phi$ if and only if it is an optimal attack for the 0/1 loss.*

*Proof.* We have:

$$\inf_{f \in \mathcal{F}(\mathcal{X})} \sup_{\mathbb{Q} \in \mathcal{A}_\varepsilon(\mathbb{P})} \mathcal{R}_{\phi, \mathbb{Q}}(f) = \mathcal{R}^\star_{\phi_\varepsilon, \mathbb{P}} = \mathcal{R}^\star_{\varepsilon, \mathbb{P}} \qquad \text{by Theorem 4.2}$$

$$= \inf_{f \in \mathcal{F}(\mathcal{X})} \sup_{\mathbb{Q} \in \mathcal{A}_\varepsilon(\mathbb{P})} \mathcal{R}_{\mathbb{Q}}(f)$$

$$= \sup_{\mathbb{Q} \in \mathcal{A}_\varepsilon(\mathbb{P})} \inf_{f \in \mathcal{F}(\mathcal{X})} \mathcal{R}_{\mathbb{Q}}(f)$$

$$= \sup_{\mathbb{Q} \in \mathcal{A}_\varepsilon(\mathbb{P})} \mathcal{R}^\star_{\mathbb{Q}}(f) = \sup_{\mathbb{Q} \in \mathcal{A}_\varepsilon(\mathbb{P})} \mathcal{R}^\star_{\phi, \mathbb{Q}} \qquad \text{by Lemma I.1}$$

$$= \sup_{\mathbb{Q} \in \mathcal{A}_\varepsilon(\mathbb{P})} \inf_{f \in \mathcal{F}(\mathcal{X})} \mathcal{R}_{\phi, \mathbb{Q}}(f)$$

$\mathbb{Q} \mapsto \inf_{f \in \mathcal{F}(\mathcal{X})} \mathcal{R}_{\phi, \mathbb{Q}}(f) = \inf_{f \in \mathcal{F}(\mathcal{X})} \mathcal{R}_{\mathbb{Q}}(f)$ is upper semi-continuous for the weak topology of measures. Moreover, $\mathcal{A}_\varepsilon(\mathbb{P})$ is compact for the weak topology of measures, then $\mathbb{Q} \mapsto \inf_{f \in \mathcal{F}(\mathcal{X})} \mathcal{R}_{\phi, \mathbb{Q}}(f)$ admits a maximum over $\mathcal{A}_\varepsilon(\mathbb{P})$. And $\mathbb{Q}$ is an optimal attack for the loss $\phi$ if and only if it is an optimal attack for the 0/1 loss. $\square$

# K  Proof of Proposition 4.2

**Proposition.** *Let us assume that $\mathcal{X}$ be a Polish space satisfying the midpoint property. Let $\varepsilon \geq 0$ and $\mathbb{P}$ be a Borel probability distribution over $\mathcal{X} \times \mathcal{Y}$. Let $\mathbb{Q}^\star$ be an optimal attack of level $\varepsilon$. Let $(f_n)_{n \in \mathbb{N}}$ be a sequence of $\mathcal{F}(\mathcal{X})$ such that $\mathcal{R}_{\phi_\varepsilon, \mathbb{P}}(f_n) \to \mathcal{R}^\star_{\phi_\varepsilon, \mathbb{P}}$. Then $\mathcal{R}_{\mathbb{Q}^\star}(f_n) \to \mathcal{R}^\star_{\varepsilon, \mathbb{P}}$.*

*Proof.* Let $(f_n)_{n \in \mathbb{N}}$ be a sequence of $\mathcal{F}(\mathcal{X})$ such that $\mathcal{R}_{\phi_\varepsilon, \mathbb{P}}(f_n) \to \mathcal{R}^\star_{\phi_\varepsilon, \mathbb{P}}$. Let $\mathbb{Q}^\star$ be an optimal attack of level $\varepsilon$. From Corollary 4.1, we get that:

$$\mathcal{R}^\star_{\phi_\varepsilon, \mathbb{P}} = \mathcal{R}^\star_{\phi, \mathbb{Q}^\star} \quad .$$

Then we get

$$0 \leq \mathcal{R}_{\phi, \mathbb{Q}^\star}(f_n) - \mathcal{R}^\star_{\phi, \mathbb{Q}^\star} \leq \mathcal{R}_{\phi_\varepsilon, \mathbb{P}}(f_n) - \mathcal{R}^\star_{\phi_\varepsilon, \mathbb{P}}$$

from which we deduce that: $\mathcal{R}_{\phi, \mathbb{Q}^\star}(f_n) \to \mathcal{R}^\star_{\phi, \mathbb{Q}^\star}$. Since $\phi$ is consistent in the standard classification setting, we then have

$$\mathcal{R}_{\mathbb{Q}^\star}(f_n) \to \mathcal{R}^\star_{\mathbb{Q}^\star} \quad .$$

$\square$

# L    About $\mathcal{H}$-calibration

Our results naturally extend to $\mathcal{H}$-calibration. With mild assumptions on $\mathcal{H}$, it is possible to recover all the results made on calibration on $\mathcal{F}(\mathcal{X})$. First, it is worth noting that, if $\mathcal{H}$ contains all constant functions, then most results about calibration in the adversarial setting extend. Proposition 2.2 naturally extends to $\mathcal{H}$-calibration as long as $\mathcal{H}$ contains all constant functions.

**Proposition.** *Let $\mathcal{H} \subset \mathcal{F}(\mathcal{X})$. Let us assume that $\mathcal{H}$ contains all constant functions. Let $\varepsilon > 0$ and $\phi$ be a continuous classification margin loss. For all $x \in \mathcal{X}$ and $\eta \in [0,1]$, we have*

$$\mathcal{C}^{\star}_{\phi_{\varepsilon},\mathcal{H}}(x,\eta) = \mathcal{C}^{\star}_{\phi,\mathcal{H}}(x,\eta) = \inf_{\alpha \in \mathbb{R}} \eta\phi(\alpha) + (1-\eta)\phi(-\alpha) = \mathcal{C}^{\star}_{\phi_{\varepsilon}}(x,\eta) = \mathcal{C}^{\star}_{\phi}(x,\eta) \quad .$$

*The last equality also holds for the adversarial $0/1$ loss.*

The proof is exactly the same as for Proposition 2.2 since we used a constant function to prove the equality. Under the same assumptions, the notion of $\mathcal{H}$-calibration and uniform $\mathcal{H}$-calibration are equivalent in the standard setting.

**Proposition.** *Let $\mathcal{H} \subset \mathcal{F}(\mathcal{X})$. Let us assume that $\mathcal{H}$ contains all constant functions. Let $\phi$ be a continuous classification margin loss. $\phi$ is uniformly $\mathcal{H}$-calibrated for standard classification if and only if $\phi$ is uniformly calibrated for standard classification. It also holds for non-uniform calibration.*

*Proof.* Let us assume that $\phi$ is a continuous classification margin loss and that $\phi$ is uniformly calibrated. Let $\xi > 0$. There exists $\delta > 0$ such that, for all $\eta \in [0,1]$, $x \in \mathcal{X}$ and $f \in \mathcal{F}(\mathcal{X})$:

$$\mathcal{C}_{\phi}(x,\eta,f) - \mathcal{C}^{\star}_{\phi}(x,\eta) \leq \delta \implies \mathcal{C}(x,\eta,f) - \mathcal{C}^{\star}(x,\eta) \leq \xi \quad .$$

Let $\eta \in [0,1]$, $x \in \mathcal{X}$ and $f \in \mathcal{H}$ such that $\mathcal{C}_{\phi}(x,\eta,f) - \mathcal{C}^{\star}_{\phi,\mathcal{H}}(x,\eta) \leq \delta$. Thanks to Proposition L, $\mathcal{C}^{\star}_{\phi,\mathcal{H}}(x,\eta) = \mathcal{C}^{\star}_{\phi}(x,\eta)$, and $f \in \mathcal{F}(\mathcal{X})$, then $\mathcal{C}_{\phi}(x,\eta,f) - \mathcal{C}^{\star}_{\phi}(x,\eta) \leq \delta$ and then:

$$\mathcal{C}(x,\eta,f) - \mathcal{C}^{\star}_{\mathcal{H}}(x,\eta) = \mathcal{C}(x,\eta,f) - \mathcal{C}^{\star}(x,\eta) \leq \xi$$

Then $\phi$ is uniformly $\mathcal{H}$-calibrated in standard classification.

Reciprocally, let us assume that $\phi$ is a continuous classification margin loss and that $\phi$ is uniformly $\mathcal{H}$-calibrated. Let $\xi > 0$. There exists $\delta > 0$ such that, for all $\eta \in [0,1]$, $x \in \mathcal{X}$ and $f \in \mathcal{H}$:

$$\mathcal{C}_{\phi}(x,\eta,f) - \mathcal{C}^{\star}_{\phi,\mathcal{H}}(x,\eta) \leq \delta \implies \mathcal{C}(x,\eta,f) - \mathcal{C}^{\star}_{\mathcal{H}}(x,\eta) \leq \xi \quad .$$

Let $\eta \in [0,1]$, $x \in \mathcal{X}$ and $f \in \mathcal{H}$ such that $\mathcal{C}_{\phi}(x,\eta,f) - \mathcal{C}^{\star}_{\phi,\mathcal{H}}(x,\eta) \leq \delta$. $\mathcal{C}_{\phi}(x,\eta,f) = \eta\phi(f(x)) + (1-\eta)\phi(-f(x))$. Let $\tilde{f} : u \mapsto f(x)$ for all $u \in \mathcal{X}$, then $\tilde{f} \in \mathcal{H}$ since $\tilde{f}$ is constant, $\mathcal{C}_{\phi}(x,\eta,f) = \mathcal{C}_{\phi}(x,\eta,\tilde{f})$ and $\mathcal{C}(x,\eta,f) = \mathcal{C}(x,\eta,\tilde{f})$. Thanks to the previous proposition, $\mathcal{C}^{\star}_{\phi,\mathcal{H}}(x,\eta) = \mathcal{C}^{\star}_{\phi}(x,\eta)$. Then: $\mathcal{C}_{\phi}(x,\eta,\tilde{f}) - \mathcal{C}^{\star}_{\phi,\mathcal{H}}(x,\eta) \leq \delta$ and then:

$$\mathcal{C}(x,\eta,f) - \mathcal{C}^{\star}_{\phi,\mathcal{H}}(x,\eta) = \mathcal{C}(x,\eta,\tilde{f}) - \mathcal{C}^{\star}_{\phi}(x,\eta) \leq \xi$$

Then $\phi$ is uniformly calibrated in standard classification. $\qquad\qquad\square$

We can now obtain the necessary and sufficient conditions as follows. They are really similar to the adversarial calibration ones.

**Proposition** (Necessary conditions for $\mathcal{H}$-Calibration of adversarial losses)**.** *Let $\varepsilon > 0$. Let $\mathcal{H} \subset \mathcal{F}(\mathcal{X})$. Let us assume that $\mathcal{H}$ contains all constant functions and that there exists $x \in \mathcal{X}$ and $(f_n)_n \in \mathcal{H}^{\mathbb{N}}$ such that $f_n(u) \to 0$ for all $u \in B_{\varepsilon}(x)$ and for all $n \in \mathbb{N}$, $\sup_{u \in B_{\varepsilon}(x)} f_n(u) > 0$ and $\inf_{u \in B_{\varepsilon}(x)} f_n(u) < 0$ Let $\phi$ be a continuous margin loss . If $\phi$ is adversarially uniformly $\mathcal{H}$-calibrated at level $\varepsilon$, then $\phi$ is uniformly calibrated in the standard classification setting and $0 \notin \operatorname{argmin}_{\alpha \in \mathbb{R}} \frac{1}{2}\phi(\alpha) + \frac{1}{2}\phi(-\alpha)$.*

**Proposition** (Sufficient conditions for $\mathcal{H}$-Calibration of adversarial losses)**.** *Let $\mathcal{H} \subset \mathcal{F}(\mathcal{X})$. Let us assume that $\mathcal{H}$ contains all constant functions. Let $\phi$ be a continuous strictly decreasing margin loss and $\varepsilon > 0$. If $\phi$ is calibrated in the standard classification setting and $0 \notin \operatorname{argmin}_{\alpha \in \mathbb{R}} \frac{1}{2}\phi(\alpha) + \frac{1}{2}\phi(-\alpha)$, then $\phi$ is adversarially uniformly $\mathcal{H}$-calibrated at level $\varepsilon$.x*

The proofs are the same as for the adversarial calibration setting. Note however that the assumptions on $\mathcal{H}$ are very weak: for instance, the set of linear classifiers

$$\mathcal{H} = \left\{ x \mapsto \langle w, x \rangle + b \mid w \in \mathbb{R}^d, b \in \mathbb{R} \right\}$$

satisfies the existence of $x \in \mathcal{X}$ and $(f_n)_n \in \mathcal{H}^{\mathbb{N}}$ such that $f_n(u) \to 0$ for all $u \in B_\varepsilon(x)$ and for all $n \in \mathbb{N}$, $\sup_{u \in B_\varepsilon(x)} f_n(u) > 0$ and $\inf_{u \in B_\varepsilon(x)} f_n(u) < 0$.