# OpenReview forum: "Towards Consistency in Adversarial Classification"
_NeurIPS.cc/2022/Conference — NeurIPS 2022 Accept_

### Official Review · Reviewer_a53P · 2022-07-11

**Rating:** 6
**Confidence:** 4
**Soundness:** 3 good
**Presentation:** 3 good
**Contribution:** 3 good

**Summary:**

This paper studies the consistency and calibration properties of adversarial losses. Specifically, it defines an adversarial loss as the highest loss in a $\epsilon$-ball around data examples and find the sufficient and necessary conditions when an adversarial loss is adversarially calibrated. Furthermore, this paper also studies the consistency of adversarial loss. However, the obtained result in Prop 4.2 regarding the consistency seems weaker than expected.

**Questions:**

1) What are $x,x',y$ in the definition of $A_\epsilon(\mathbb{P})$?
2) In Cor 4.2 and Prop 4.2, does $\phi$ needs to be a 0/1 like loss?
3) Why does not $\epsilon$ exist in the sufficient conditions in Theorem 3.2? Does it mean that the conclusion holds for every $\epsilon$?

**Limitations:**

This is a theoretically comprehensive study. However, it would be much better if the authors give hints of how the gained results can be exploited to benefit adversarial attack and defense.

**Strengths And Weaknesses:**

* Strengths:
- This is a comprehensive study about the calibration and consistency property of loss functions in the context of adversary.
- The paper is generally well-written and easy to follow.

* Weakness:
- The gained results seem to have less impact on practical problems of adversarial attack and defense.
- There is no hint of how to utilize the gained results to improve adversarial robustness.

---

> ### Author Response · Authors · 2022-07-29
> **Thank you!**
>
>
> We would like to thank the reviewer for their time and
> appreciation of our paper.
>
> **On the practical impact.**  We agree that the practical impact is limited so far. There is still a gap between practice and theory. We hope that this paper lead to rethink the usage of losses in practice in machine learning. As pointed out in the response to reviewer MTmv, we believe that it may still be a bit premature to start experimenting in this research area, especially because we do not provide strong consistency results yet.
>
> Nevertheless, we believe that our work could have a concrete practical impact in the near future, especially when considering adversarial training [B]. When implementing this defense, we are essentially optimizing an surrogate adversarial loss. As our results show, these losses are not consistent when chosen with standard consistency theory, which can lead to a large optimality gap when using these techniques. We believe that shifted odd losses could circumvent this problem, which would give important practical significance to our work. We will discuss this point in the conclusion of the paper.
>
>
> [B] Madry, A., Makelov, A., Schmidt, L., Tsipras, D., and Vladu, A. Towards deep learning models resistant to adversarial attacks. International Conference on Learning Representations, 2018.
>
>
> **On Definition 4.4.** $(x,y)$ denotes the input-label pair from the original distribution while $(x',y')$ denotes the input-label pair from the perturbed distribution (after attack). We will make it clearer in the updated version.
>
>
> **On Corollary 4.2 and Proposition 4.2.** Yes indeed $\phi$ need to be a $0/1$ like loss, we will add it to the revision.
>
> **On the validity of Theorem 3.2.** Yes indeed, the result holds for every $\varepsilon$, we will make that clear.

---

> > ### Comment · Reviewer_a53P · 2022-08-08
> > **Thanks for answering my questions**
> >
> > Thanks for addressing my questions. It would be great if this research comes with stronger practical evidences. I decide to increase my score.

---

### Official Review · Reviewer_cdXZ · 2022-07-11

**Rating:** 9
**Confidence:** 3
**Soundness:** 3 good
**Presentation:** 4 excellent
**Contribution:** 3 good

**Summary:**

The paper shows that the usual continous loss functions like hinge loss and logistic loss that are used to approximate the $0/1$-loss are not consistent nor calibrated for the adversarial $0/1$-loss.

It shows that a certain assymetry is necessary in order to have calibration and proposes shifted odd functions as candidates that are shown to fullfill them.

Finally, difficulties with and potential pathways towards consistent losses for the adversarial setting are discussed and related with some interesting theoretical results.

The paper does not contain any experiments.

**Questions:**

The definition/interpretation of $\eta$ should be clarified. In Definition 2.4, it is the conditional probability of the ground truth being $1$ for a given input $x$. While it is stated here that this is the meaning "in this case", it is not defined otherwise in the rest of the paper.
It seems like $\eta$ is a function of $x$? If so, it would make things clearer if it was written as $\eta(x)$ or $\eta_x$. If it isn't, pleased clarify.
Since the ground truth may not be constant in the $\epsilon$-ball around a given $x$, it appears to be a problem that the statements and proofs for the adversarial settings seem to use one constant $\eta$ over the ball. For example, in Appendix E in the equation of line 512, $\eta$ appears outside of the supremum over $x'$.


In Definition 4.4, $x,x'$ and $y,y'$ should be defined more explictly. It would help to include an explanation of the definition, for which referring to "[12] Meunier et al.: Mixed Nash Equilibria in the Adversarial Examples Game" didn't help me either.

In l. 139, I think there is a sign error in the ramp loss, right?

In l. 354, is "anticipated" the correct word?

A question about alternative losses: Would using logistic loss (KL-divergence) for evaluating adversarially robust models be an interesting metric that would lead to fewer problems since one does not need a surrogate to train it?

A comment and potentially a discussion in the revision on the relation to the very recent "Frank et al.: The Consistency of Adversarial Training for Binary Classification (https://arxiv.org/abs/2206.09099) " which at first glance states some contradicting results would be very welcome.

**Limitations:**

The limitations of currently achievable results on consistency are described in detail.

The authors "believe there is no potential social harm" which is not a statement that is not obvious for any machine learning research, due to potential dual use and other general risks.
However, there is nothing specific that needs to be mentioned for this paper, and I agree with the authors as in my opinion a work like this has non-negative expected social impacts.

**Strengths And Weaknesses:**

The paper is generally well written and presented.

Showing that the standard losses for adversarial robustness training do not satisfy certain basic theoretical properties is an interesting and potentially important insight. The paper rigorously and comprehensively proofs these results.

I checked the proofs up until Proposition 3.1 and they appear to be correct, with the only issue for me being the definition of $\eta$, see below.

Figure 1 is very helpful for understanding Proposition 3.1; more similar illustrations for the other results would be very beneficial for the paper, if possible.

I think that the paper is a very interesting contribution to the field of adversarial robustness.
The paper is generally well written but some details are missing for me (also due to me not being very familiar with the literature on consistency); I hope for clarifications on the questions below, particularly on $\eta$.

While the results are interesting and partly surprising to me from the perspective of adversarial robustness, I cannot say much about the novelty in the context of consistency and calibration literature which this work builds upon.

---

> ### Author Response · Authors · 2022-07-29
> **Thank you! (Part 1/2)**
>
> We would like to thank the reviewers for their appreciation of our paper and their valuable insights.
>
> **On the suggestion to add figures.** We will add one or more figures to explain why some usual losses are not calibrated or to illustrate the counterexamples we provide. If the reviewer would like to make additional suggestions that would be helpful to the understanding of the article, we will be happy to incorporate them.
>
>
> **On the interpretation of $\\eta$.** In definition 2.4, $\\eta$ is a dummy variable. In fact the calibration function is defined as  $C_L: (x,\\eta,f)\\mapsto \\eta L(x,1,f) +(1-\\eta) L(x,-1,f)$. Accordingly, $\\eta$ as presented in definition 2.4, is *not necessarily* the conditional distribution of $y=1$ knowing $x$. However, what we meant to say in the discussion below definition 2.4 is that, for any $x \\in \\mathcal{X}$ and any $\\eta \\in [0,1]$, there exist a distribution $\\mathbb{P}:= \\eta \\delta_{(x,1)} + (1-\\eta) \\delta_{(x,-1)}$ such that for any $f\\in\\mathcal{F}(\\mathcal{X})$, we have $C_L(x,\\eta,f)=\\mathcal{R}_{L,\\mathbb{P}}(f)$.  This way of re-writing the risk associated to a loss $L$ with a distribution that depends both on $x$ and $\\eta$ is particularly useful to study calibration and applies to any loss (including adversarial losses). When we match the calibration function with the risk of distribution $\\mathbb{P}$ in that way, we indeed have $\\eta=\\mathbb{P}(y=1 \mid x)$; hence $\\eta$ is dependent on $x$. While $\\eta_x$ or $\\eta(x)$ would have been valid notations, we chose to keep the notation $\\eta$ to follow the convention established in [A]. We will include this discussion in the final version to provide a better insight on what $\\eta$ represents and its dependency to $x$.
>
> [A] Peter L Bartlett, Michael I Jordan, and Jon D Mc Auliffe. Convexity, classification, and risk bounds. Journal of the American Statistical Association, 101(473):138–156, 2006.
>
> **On Definition 4.4.** $(x,y)$ denotes the input-label pair from the original distribution while $(x',y')$ denotes the input-label pair from the perturbed distribution (after attack). We will make it clearer in the updated version.
>
> **On the ramp loss.** There is indeed a typo. The ramp loss is $\phi(t) = \max(1-t, 0) - \max (-t-1, 0) $. We will correct it accordingly. Thank you for spotting this.
>
> **On the use of logistic loss for evaluating adversarially robust models.**  In classification settings, the final evaluation of models is often made on the accuracy (i.e. 0/1 loss), that is why we studied calibration wrt the 0/1 loss. If the metric of interest was indeed the cross-entropy, our study would not readily apply. It is however an interesting future work to change the classification objective and understand the impact of this change on the adversarial setting.

---

> > ### Comment · Reviewer_cdXZ · 2022-08-09
> > **Re: Authors' response**
> >
> > Thanks for the clarifications!
> >
> > After reading the responses and checking a few more references, I have raised my score from 8 to 9, since I think the paper can be an important contribution towards understanding which adversarial robustness requirements can and cannot be successfully optimized.
> >
> > I'm still unsure about the question of the dependence of $\eta$ on $x'$ within the ball, and unsure on whether this question is crucial for the whole derivation.
> > The authors clarified in the above response that $\eta$ depends on $x$, and how it can be interpreted.
> > However, the following questions remain still open for me:
> >  - The stated dependence on $\eta$ on $x$ also means that it cannot be assumed to be constant on $B_\epsilon(x)$, right?
> >  - Then, since $\eta$ depends on $x' \in B_\epsilon(x)$, having it outside of the $sup_x' \in B_\epsilon(x)$ in the equation after line 512 cannot be correct, or is it?
> >  - In case this is indeed a problem, what is its impact and can it be fixed?

---

> > > ### Author Response · Authors · 2022-08-09
> > > **Thanks!**
> > >
> > > Thank your for the response you gave!
> > >
> > > We are not sure to understand your point. Assuming that $\eta$ depends on $x$ (denoted $\eta(x)$) then the calibration function is defined as: $C_ {\phi_ \varepsilon} (x,\eta(x),f) = \eta(x)\sup_ {x'\in B_ \varepsilon(x)} \phi(f(x')) + (1-\eta(x)) \sup_{x'\in B_\varepsilon(x)} \phi(-f(x'))$ . Then $\eta$ only depends on $x$ but not on $x'$. So in Eq L512, the equation seems to be true.
> > >
> > > Do not hesitate to tell if it answer your question!

---

> ### Author Response · Authors · 2022-07-29
> **Thank you (Part 2/2)**
>
>
>
> **Comparison with ``[C] The Consistency of Adversarial Training for Binary Classification, Natalie S. Frank, Jonathan Niles-Weed''**
>
> We thank the reviewer for pushing the reference (the reference was arxived after the submission). The results seem indeed very promising. However, we are not confident with the technical soundness of the paper, as we think we have found a counterexample to their main theorem. We detail the counter-example below.
>
>
> Let $\\mathcal{X}=\\mathbb{R}$, $\\varepsilon=1$, $\\mathbb{P}(\cdot|y=\pm 1) = \\mathcal{U}([0,1])$, $\\mathbb{P}(y=\pm1)=\frac12$ and $\\phi$ a continuous decreasing loss such that $0\\in\\arg\\min \\phi(\\alpha)+\\phi(-\\alpha)$ (for instance the logistic or Hinge loss which are consistent). Assumptions 1 and 2 of [C] are then satisfied. Then, for all functions $f \\in \\mathcal{F}(\\mathcal{X})$, the following holds: $\\mathcal{R}_ {\\phi_\\varepsilon,\\mathbb{P}\}(f)\\geq \\mathcal{R}_{\\phi,\\mathbb{P}}(f) = \\frac12 \\int \\phi(f(x))+ \\phi(-f(x)) dx \\geq \\frac12 \\int 2 \\phi(0) dx =\\phi(0)$
>
> Furthermore, taking $f^*:x\\mapsto 0$, we deduce that $\\mathcal{R}_ {\\phi_\\varepsilon,\\mathbb{P}}(f^*) = \\phi(0) = \\mathcal{R}_ {\\phi_\\varepsilon,\\mathbb{P}}^\star$. Note also that by definition of $\mathbb{P}$, we have $\mathcal{R}_ {\varepsilon,\mathbb{P}}^\star = \frac12$.
>
> Now let us take a sequence $(f_ n)_ n \in \mathcal{F}(\mathcal{X})^{\mathbb{N}}$ such that for any $n \in \mathbb{N}$, we have $f_n : x\mapsto \frac1n$ if $x>\frac12$ and $-\frac{1}{n}$ otherwise. With this construction, for every $n \in \mathbb{N}$ and $x\in [0,1]$, we have $\\sup_ {x'\in [x\pm1]} \\phi(f_n(x'))=\\sup_ {x'\in [x\pm1]} \\phi(-f_n(x')) = \\phi(\frac1n).$
>
>
> Hence $\\mathcal{R}_ {\\phi_\\varepsilon,\\mathbb{P}}(f_n)= \\phi(\\frac1n)\to \phi(0) = \\mathcal{R}_ {\phi_\varepsilon,\mathbb{P}}^\star$ as $n\to\infty$. Moreover we have $\sup_ {x'\in [x\pm1]} \mathbf{1}_ {f_n(x')\leq 0}=\sup_ {x'\in [x\pm1]} \mathbf{1}_ {f_n(x')> 0}= 1 .$
>
> Accordingly $\mathcal{R}_ {\varepsilon,\mathbb{P}}(f_n) =1\not\to \frac12 = \mathcal{R}_ {\varepsilon,\mathbb{P}}^\star$. As we just build a sequence of function $(f_ n)_ n$ for which $\mathcal{R}_ {\phi_\varepsilon,\mathbb{P}}(f_n) \to \mathcal{R}_ {\phi_\varepsilon,\mathbb{P}}^\star$ but $\mathcal{R}_ {\varepsilon,\mathbb{P}}(f_n) \not\to \mathcal{R}_ {\varepsilon,\mathbb{P}}^\star$, $\phi$ cannot be consistent.
>
> We believe the assumption that $0\not\in\arg\min \phi(\alpha)+\phi(-\alpha)$ is essential in the consistency study and it has not been taken into account in their assumptions. Nevertheless, we may have missed a point since we did not sufficiently read their paper to be confident enough about the results given the assumptions the authors provide. We will try to assess carefully the validity of the proofs and we will contact the authors of the paper to discuss their results. Thanks again for pointing this paper out.

---

### Official Review · Reviewer_MTmv · 2022-07-16

**Rating:** 7
**Confidence:** 3
**Soundness:** 3 good
**Presentation:** 3 good
**Contribution:** 3 good

**Summary:**

This paper studies the problem of consistency and calibration in the setting of classification with an adversary that perturbs the inputs during inference. The authors highlight the critical difference in the consistency and calibration between the standard setting and the adversarial setting. They show the widely used convex surrogate losses for 0-1 loss are no longer calibrated or consistent. They then provide detailed analyses to build the necessary and sufficient conditions for continuous loss functions to be calibrated in an adversarial setting. They further advance toward consistency in the adversarial setting by showing a weak result of consistency.

**Questions:**

Is it possible to analyze the level of “divergence” of widely used loss functions such as the logistic loss from the 0-1 loss? And perhaps provide some empirical results to validate it. Such analyses would be extremely helpful to motivate people to rethink the loss function for adversarial training and potentially provide new insights.

**Limitations:**

The limitations of this paper have been adequately discussed at the end of Section 4.2.

**Strengths And Weaknesses:**

The originality of the work is sufficient as the studies of calibration and consistency in adversarial settings for general functions are missing in the literature to the best of my knowledge.

The quality of the work is high as the authors provide detailed and solid analyses of calibration and consistency in the adversarial setting, including the difficulty and (potential) solutions to specific and general results.

The paper is well-organized and clarified, with sufficient background and preliminaries provided. I am not an expert in calibration and consistency but I was able to follow the analyses step by step.

Finally, I believe the result is significant as the calibration and consistency in adversarial settings are important problems that are able to guide the training of adversarially robust classifiers. This paper is able to move one step toward this goal.

---

> ### Author Response · Authors · 2022-07-29
> **Thank you!**
>
>
> We would like to thank the reviewer for their time and appreciation of our paper.
>
> We agree that an empirical analysis would be useful to "quantify" the non-consistency of standard losses such as the cross-entropy loss. However, we believe that such an experiment could be misleading because (i) we do not have yet strong consistency results and (ii) it is difficult to replicate the theoretical framework, as we cannot optimize classifiers on the space of measurable functions and the distributions are empirical (i.e., our analysis does not account for the generalization gap). Thus, we believe that it may still be a bit premature to start experimenting in this research area.
>
> Nevertheless, we believe that our work could have a concrete practical impact in the near future. In fact, when we try to defend a model by training it adversarially (e.g., using adversarial training [B]), we are actually optimizing a surrogate adversarial loss. As our results show, these losses may not necessarily consistent in the adversarial setting. As pointed out by our theoretical findings, using these techniques may lead to a large optimality gap. From our analysis, we believe that shifted odd losses could circumvent this problem, which would give important practical significance to our work. We will return to this point in the conclusion of this paper.
>
>
> [B] Madry, A., Makelov, A., Schmidt, L., Tsipras, D., and Vladu, A. Towards deep learning models resistant to adversarial attacks. International Conference on Learning Representations, 2018.

---

### Meta-Review · Area_Chair_9Xwi · 2022-08-26

**Recommendation:** Accept
**Confidence:** Certain

**Metareview:**

Reviewers have expressed strongly in favour of acceptance, one improving their score to very strong accept after the rebuttal and discussion. Congratulations! I’m delighted to recommend acceptance.

**Award:**

No

---

### Decision · Program_Chairs · 2022-09-14

Accept